# R&B: Region and Boundary Aware Zero-shot Grounded Text-to-image Generation

**Jiayu Xiao**[1,2][*]  **Henglei Lv**[1,2][*]  **Liang Li**[1] [†]  **Shuhui Wang**[1,3]  **Qingming Huang**[1,2,3]

[1]Key Lab of Intelligent Information Processing, ICT, CAS, Beijing, China

[2] University of Chinese Academy of Sciences, Beijing, China

[3] Peng Cheng Laboratory, Shenzhen, China

`jiayu.xiao@vipl.ict.ac.cn`  `liang.li@ict.ac.cn`  `qmhuang@ucas.ac.cn`

## Abstract

Recent text-to-image (T2I) diffusion models have achieved remarkable progress in generating high-quality images given text-prompts as input. However, these models fail to convey appropriate spatial composition specified by a layout instruction. In this work, we probe into zero-shot grounded T2I generation with diffusion models, that is, generating images corresponding to the input layout information without training auxiliary modules or finetuning diffusion models. We propose a **R**egion and **B**oundary (R&B) aware cross-attention guidance approach that gradually modulates the attention maps of diffusion model during generative process, and assists the model to synthesize images (1) with high fidelity, (2) highly compatible with textual input, and (3) interpreting layout instructions accurately. Specifically, we leverage the discrete sampling to bridge the gap between consecutive attention maps and discrete layout constraints, and design a region-aware loss to refine the generative layout during diffusion process. We further propose a boundary-aware loss to strengthen object discriminability within the corresponding regions. Experimental results show that our method outperforms existing state-of-the-art zero-shot grounded T2I generation methods by a large margin both qualitatively and quantitatively on several benchmarks. Project page: https://sagileo.github.io/Region-and-Boundary.

## 1 Introduction

Recently, text-to-image (T2I) models such as DALL-E 2 (Ramesh et al., 2022), Imagen (Saharia et al., 2022), and Stable Diffusion (Rombach et al., 2022) have shown remarkable capacity of synthesizing high-quality images conditioned on textual input. Trained on large-scale datasets (Schuhmann et al., 2022; Ramesh et al., 2021; Lin et al., 2014) composed of image-text pairs collected from the Internet, these models learn a coherent generative space where vision and language modalities are highly unified. Many works (Ruiz et al., 2023; Zhang & Agrawala, 2023; Hertz et al., 2022; Wu et al., 2022) leverage strong generative prior and zero-shot capacity of these T2I models and extend them to various downstream tasks, presenting diverse personalized generative results based on user instructions.

Nevertheless, there remains challenges for grounded T2I generation. As shown in Figure 1, in some situations, users may expect outcome that the generated images are not only highly relevant to the given prompt but also adhere to a specified layout. Existing T2I models struggle (Gokhale et al., 2022) to correctly interpret spatial layout instructions specified by textual input, and lack the ability to localize concepts according to other spatial conditional inputs (*e.g.*, bounding boxes). To address this, some works (Zheng et al., 2023; Li et al., 2023a; Avrahami et al., 2023) propose to train plug-in modules to incorporate layout conditions into the generative process. Yet this requires auxiliary training process on labelled data, and does not guarantee generalization on new model architectures.

Most recent works (Chen et al., 2023; Phung et al., 2023; Epstein et al., 2023; Couairon et al., 2023; Mou et al., 2023; Xie et al., 2023) show that self-attention and cross-attention maps of diffusion

---

[†]Corresponding Author.
[*]Equal contribution.

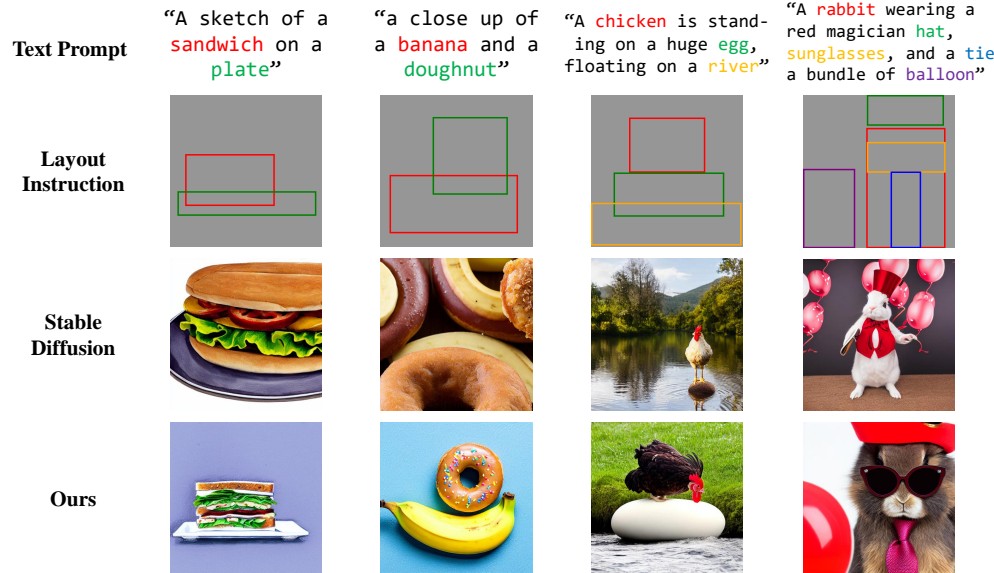

Figure 1: Illustration of zero-shot grounded text-to-image generation with Stable Diffusion. Given text input and bounding box information corresponding to the specific concepts, our method can generate images that conform to the spatial instructions.

models encode rich structural information (Hertz et al., 2022), and manipulation on the attention maps leads to edited results on the corresponding images. Inspired by the classifier guidance (Dhariwal & Nichol, 2021), they design energy functions on the attention maps for zero-shot layout control by transforming spatial guidance signals into gradients. While showing competitive results that interpret the layout instructions, they still suffer from two critical issues. First, these methods fail to provide accurate spatial guidance, reflected by the misalignment between the synthetic images and the layout information. Second, they inherit the semantic inconsistency issues (*e.g.*, missing objects, conceptual binding errors) from the original T2I model.

To mitigate the above issues, we propose **R**egion and **B**oundary (R&B) aware cross-attention guidance for grounded T2I generation. As for region-aware guidance, we find that the inaccurate outputs of previous methods owes to the neglect of the gap between cross-attention maps and ground-truth bounding boxes. The former depicts shape and localization of objects at a fine-grained level, while the latter only provides coarse-grained spatial clues. To address this, we first leverage a dynamic threshold to highlight object-related regions of cross-attention maps, and extend them to their minimum bounding rectangles (MBR). Then, inspired by theories on discrete sampling (Jang et al., 2016; Sohn et al., 2015) for deep networks, we construct a differentiable path from the binary MBRs to consecutive attention maps through a Straight-Through Estimator. Further, we design a region-aware loss that directly measures divergence between MBRs and ground-truth boxes, enforcing alignment between the distribution of attention maps and layout conditions. As for boundary-aware guidance, we discover that refining boundaries of attention maps in object regions enhances the semantic consistency to the textual prompts, and further ensures adherence to layout constraints. We then propose a boundary-aware loss to sharpen boundaries of attention maps to encourage expression of different concepts in correct regions and enhance faithfulness of the synthetic images.

Our contribution can be concluded as below:

1. To achieve zero-shot grounded T2I generation for diffusion models, we introduce R&B, a novel attention guidance approach for layout generation, which requires no auxiliary modules or extra training.

2. We perform attention control of different objects from the perspectives of region and boundary respectively. Accordingly, we design a region-aware loss and a boundary-aware loss. The former encourages accurate alignment between activation regions of cross-attention maps and layout instructions, while the latter enhances expressiveness of different concepts in the refined localization, and better cooperates with text semantics.

3. We conduct experiments and ablation study for better comprehension of the proposed method. Experimental results show that our proposed R&B performs well on zero-shot grounded T2I generation with high spatial accuracy and generative fidelity, and surpasses the existing state-of-the-art methods by a large margin both qualitatively and quantitatively.

## 2 BACKGROUND

**Diffusion models.** A diffusion model is trained to approximate a data distribution via gradually denoising a random variable drawn from a unit Gaussian prior. Essentially, it learns to reverse a time-dependent data perturbation process that diffuses the data distribution into a fixed prior distribution, *e.g.*, Gaussian distribution. Basically, a denoiser $\epsilon_\theta$ is trained to estimate the noise $\epsilon_t$ from the corrupted image $z_t = \alpha_t x + \sigma_t \epsilon_t$ with the diffusion loss:

$$L(\theta) = \mathbb{E}_{x_0, t, \epsilon_t \sim \mathcal{N}(0,1)}[||\epsilon_t - \epsilon_\theta(z_t, t)||^2], \qquad (1)$$

Recently, a powerful Stable Diffusion (SD) (Rombach et al., 2022) is widely used for synthesizing images with high resolution and quality. It transforms real images into latents and performs denoising process in the latent space. It further introduces a fixed text encoder (Radford et al., 2021) for text-image interactions with cross-attention modules to achieve text conditional image generation.

**Controllable diffusion.** Previous work (Song et al., 2020b) describes diffusion models as score-based models from the perspective of stochastic differential equation (SDE), and transforms the derivation of the reversed diffusion process into estimation of the score function: $\nabla_{z_t} \log p_t(z_t)$. Further, the controllable diffusion process can be viewed as a composed score function to sample from a richer distribution, *i.e.*, $\nabla_{z_t} \log p_t(z_t, y)$, where $y$ is an external condition. This joint score function can be decomposed as below:

$$\nabla_{z_t} \log p_t(z_t, y) = \nabla_{z_t} \log p_t(z_t) + \nabla_{z_t} \log p_t(y|z_t), \qquad (2)$$

where the first term is the original unconditional model, and the second term corresponds to the classifier guidance that steers the sampling process towards the generation target (*e.g.*, categories). Dozens of methods (Voynov et al., 2023; Bansal et al., 2023; Chen et al., 2023; Mou et al., 2023; Lv et al., 2024) design different energy functions to estimate $\nabla_{z_t} \log p_t(y|z_t)$ for controllable generation. Inspired by these works, we incorporate layout conditions into the denoising process through an energy function $g(z_t; t, y)$, where lower value of $g$ indicates better alignment with spatial constraints. In this way, the guidance $\nabla_{z_t} \log p_t(y|z_t)$ can be computed as $\eta_g \nabla_{z_t} g(z_t; t, y)$, where $\eta_g$ is a scaling factor to control the guidance strength.

**Text-to-image generation with layout guidance.** Many recent works inject layout guidance into the powerful SD models and achieve interactive generation that satisfies users' preference. Some of them (Zhang & Agrawala, 2023; Li et al., 2023a) train auxiliary modules to embed the layout information into intermediate features of the SD UNet to bias model output. Other works provide a solution of cross-attention (Tu et al., 2023b;a; 2024) modulation to inject the layout guidance in either forward or backward manner. The forward methods (Liu et al., 2023b; Kim et al., 2023; Balaji et al., 2022) directly manipulate the value of the SD cross-attention maps during the forward process, requiring more refined layout information (*e.g.*, segmentation maps) or additional training. The backward methods (Chen et al., 2023; Xie et al., 2023; Phung et al., 2023) utilize an energy function to transform layout constraints into gradients and update the noisy latents during the sampling process. Our proposed R&B can be viewed as a backward method.

## 3 METHOD

In this section, we first provide a detailed description of the problem definition of zero-shot grounded text-to-image generation. We then discuss the methodology of our R&B for diffusion-based zero-shot grounded T2I generation. Figure 2 illustrates an overview of our training-free generation framework.

**Zero-shot grounded T2I generation.** In contrast to visual grounding (Liu et al., 2022), zero-shot grounded text-to-image generation (Chen et al., 2023; Phung et al., 2023; Xie et al., 2023; Lian et al.,

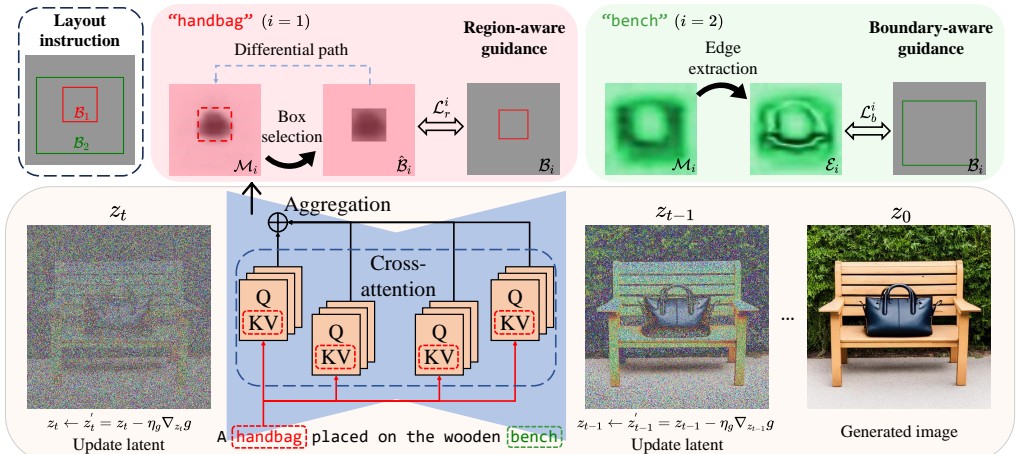

Figure 2: Overall framework of our region-aware cross-attention guidance for diffusion-based zero-shot grounded T2I generation. At each time step, we update the noisy latent via optimizing the energy function $g(z_t; t, \mathbf{B})$, which is a summation of region-aware loss $\mathcal{L}_r$ and boundary-aware loss $\mathcal{L}_b$.

2023) aims at generating images conditioned on a text prompt and grounding information, without additional training procedure. Considering a text prompt $\mathbf{T} = \{\mathcal{T}_1, ..., \mathcal{T}_N\}$, and a set of bounding boxes $\mathbf{B} = \{\mathcal{B}_1, ..., \mathcal{B}_K\}$. Each bounding box $\mathcal{B}_i$ corresponds to a subset $\mathbf{T}_i \subset \mathbf{T}$. The generated images are supposed to be highly consistent with the text prompt and satisfy the layout constraint imposed by bounding boxes. An intuitive visual example is shown in Figure 2, given the textual input "A handbag placed on the wooden bench" and layout instruction with two bounding boxes associated with "handbag" and "bench" respectively, our approach synthesizes an image in which a handbag is positioned in the middle of a wooden bench, and the object sizes of the handbag and bench match the two corresponding bounding boxes.

**Attention map extraction and aggregation.** As shown in Figure 2, at each denoising step $t$, the noisy latent $z_t$ is fed to the diffusion U-Net. At each cross-attention layer $l$, the image and text features are fused for cross-modal interaction, where deep image features $\phi_l(z_t)$ and textual embeddings are projected to query matrix $Q_l$ and key matrix $K_l$ respectively (here we disregard the case of multi-head attention for simplicity). The cross-attention $\mathcal{A}_l \in \mathbb{R}^{H_l \times W_l \times N}$ with respect to $N$ textual tokens at layer $l$ is computed as below:

$$\mathcal{A}_l = \text{Softmax}(\frac{Q_l K_l^\top}{\sqrt{d}}), \tag{3}$$

where the dimension $d$ is utilized to normalize the softmax values. $H_l$ and $W_l$ represent the resolution of the cross-attention maps. Denoting $\{\mathcal{A}_l^j \in \mathbb{R}^{H_l \times W_l} | \mathcal{T}_j \in \mathbf{T}_i\}$ as the cross-attention maps with respect to the $i^{\text{th}}$ concept $\mathbf{T}_i$, we compute an aggregated attention map (Couairon et al., 2023) $\mathcal{M}_i$ by averaging the cross-attention maps across different layers:

$$\mathcal{M}_i = \frac{1}{L} \sum_{l=1}^{L} \sum_{j=1}^{N} \mathbb{1}_{\mathbf{T}_i}(\mathcal{T}_j) \cdot \mathcal{A}_l^j, \tag{4}$$

where $\mathbb{1}_{\mathbf{T}_i}(\cdot)$ is an indicator function. Since the cross-attention maps $\mathcal{A}_l$ at each layers have different resolutions, we first upsample them to an unified resolution $64 \times 64$, then average them to get $\mathcal{M}_i$. Previous works (Hertz et al., 2022; Epstein et al., 2023; Tang et al., 2023) indicate that cross-attention maps of the diffusion U-Net contain rich semantic and geometric information for generated images. Based on the above properties, we perform cross-attention guidance for grounded generation.

**Dynamic thresholding.** After adopting the aggregated cross-attention maps, our goal is to accurately describe the shape and localization of a specific concept in a zero-shot manner. Previous work (Epstein et al., 2023) applies a fixed threshold to the normalized attention map to highlight the foreground objects, and performs self-guided image editing. However, this approach is ineffective in the case of grounded generation, because the distribution of the attention map changes during the layout

generation process. To address this, we propose a dynamic thresholding approach to highlight the foreground region $\hat{\mathcal{M}}_i$ of the $i^{\text{th}}$ object:

$$\mathcal{M}_i^{\text{norm}} = \frac{\mathcal{M}_i - \min_{h,w}(\mathcal{M}_i)}{\max_{h,w}(\mathcal{M}_i) - \min_{h,w}(\mathcal{M}_i)}, \tag{5}$$

$$\tau_i = \lambda \frac{\sum_{h,w} \mathcal{M}_i^{\text{norm}} \odot \mathcal{B}_i}{\sum_{h,w} \mathcal{B}_i} + (1-\lambda)\frac{\sum_{h,w} \mathcal{M}_i^{\text{norm}} \odot (1-\mathcal{B}_i)}{\sum_{h,w} 1 - \mathcal{B}_i}, \tag{6}$$

$$(\hat{\mathcal{M}}_i)_{h,w} = \begin{cases} 1 & \text{if } (\mathcal{M}_i^{\text{norm}})_{h,w} \geq \tau_i \\ 0 & \text{if } (\mathcal{M}_i^{\text{norm}})_{h,w} < \tau_i \end{cases}, \tag{7}$$

where $\tau_i$ is calculated by weighted averaging the activation values inside and outside the corresponding bounding box $\mathcal{B}_i$ in Eq. (6). Tuple $(h, w)$ represents a spatial entry of the attention map.

**Box selection.** The binary mask $\hat{\mathcal{M}}_i$ in Eq. (7) highlights the foreground region associated with the $i^{\text{th}}$ concept. Nonetheless, there remains a domain gap (Cai et al., 2023; Xiao et al., 2022; Liu et al., 2023a) between $\hat{\mathcal{M}}_i$ and the ground-truth bounding box $\mathcal{B}_i$, since the latter only describes the shape, size, and position of the object through a rectangular binary mask. To bridge the gap, we perform box selection by extending $\hat{\mathcal{M}}_i$ to its minimum bounding rectangle (MBR) $\hat{\mathcal{B}}_i$ to match $\mathcal{B}_i$, as illustrated in Figure 2. Even though, directly optimizing $\hat{\mathcal{B}}_i$ towards its ground-truth $\mathcal{B}_i$ is not feasible because the MBRs are hard and non-differentiable masks. Towards practical implementation, we first adopt two variants of $\mathcal{M}_i$: a shape-aware attention map $\mathcal{M}_i^s$ (Epstein et al., 2023) and an appearance-aware attention map $\mathcal{M}_i^a$ (see Appendix C for details). Then we construct a differentiable path from the hard mask $\hat{\mathcal{B}}_i$ to the two consecutive attention maps via a Straight-Through Estimator (STE) for the region-aware cross-attention guidance:

$$\hat{\mathcal{B}}_i^s = \texttt{stopgrad}(\hat{\mathcal{B}}_i - \mathcal{M}_i^s) + \mathcal{M}_i^s, \tag{8}$$

$$\hat{\mathcal{B}}_i^a = \texttt{stopgrad}(\hat{\mathcal{B}}_i - \mathcal{M}_i^a) + \mathcal{M}_i^a. \tag{9}$$

Numerically, the two binary masks $\hat{\mathcal{B}}_i^s$ and $\hat{\mathcal{B}}_i^a$ in Eqs. (8) and (9) equal to $\hat{\mathcal{B}}_i$. However, since $\hat{\mathcal{B}}_i^s$ and $\hat{\mathcal{B}}_i^a$ are differentiable, we can directly design loss functions on the above two binary masks to transform the layout condition into gradients, and optimize the input noisy latent through error backpropagation for layout control.

**Region-aware loss for attention guidance.** After obtaining the aforementioned differentiable boxes, we align them with the ground-truth bounding box $\mathcal{B}_i$ through a region-aware loss:

$$\text{IoU}_i = \frac{\sum_{h,w} \hat{\mathcal{B}}_i \odot \mathcal{B}_i}{\sum_{h,w} \hat{\mathcal{B}}_i + (1 - \hat{\mathcal{B}}_i) \odot \mathcal{B}_i}, \tag{10}$$

$$\mathcal{L}_r^i(z_t; \mathcal{B}_i) = (1 - \text{IoU}_i) \cdot (\lambda_s(1 - \frac{\sum_{h,w} \hat{\mathcal{B}}_i^s \odot \mathcal{B}_i}{\sum_{h,w} \hat{\mathcal{B}}_i^s}) + \lambda_a(1 - \frac{\sum_{h,w} \hat{\mathcal{B}}_i^a \odot \mathcal{B}_i}{\sum_{h,w} \hat{\mathcal{B}}_i^a})). \tag{11}$$

The fraction in $\mathcal{L}_r^i$ encourages the attention map to transport its top values from current high-activation regions into the target regions. $\text{IoU}_i$ in Eq. (10) measures the spatial accuracy of generative layout, and controls the scale of $\mathcal{L}_r^i$. When $\hat{\mathcal{B}}_i$ completely agrees with $\mathcal{B}_i$, $\mathcal{L}_r^i$ falls to 0. Different from energy functions proposed in previous methods (Phung et al., 2023; Chen et al., 2023; Xie et al., 2023), $\mathcal{L}_r^i$ in Eq. (11) directly measures the divergence between the predicted MBR and ground-truth $\mathcal{B}_i$, which provides a more accurate guidance path for the diffusion model, leading to better generation accuracy in accordance with layout constraints.

**Boundary-aware loss for attention guidance.** Recent work (Li et al., 2023b) proposes to enhance the faithfulness of synthetic images by incentivizing the presence of objects via maximizing the total variation of cross-attention maps, which promotes local changes in attention maps and facilitates the appearance of discriminative object-relative regions within the image. Inspired by this, we propose a boundary-aware loss to enlarge the variation of cross-attention maps within the regions corresponded to different objects:

$$\mathcal{E}_i = \texttt{Sobel}(\mathcal{M}_i), \tag{12}$$

$$\mathcal{L}_b^i(z_t; \mathcal{B}_i) = (1 - \text{IoU}_i) \cdot (1 - \frac{\sum_{h,w} \mathcal{E}_i \odot \mathcal{B}_i}{\sum_{h,w} \mathcal{E}_i}), \tag{13}$$

where $\texttt{Sobel}(\cdot)$ represents the Sobel Operator and $\mathcal{E}_i$ is the extracted edge map from the aggregated attention map. Intuitively, $\mathcal{L}_b^i$ enlarges the sum of changes in $\mathcal{B}_i$ and suppresses the outer activation, thus constrains the objects to express within bounding boxes and better adheres to the semantics. To this way, the overall energy function for classifier guidance in Section 2 can be written as:

$$g(z_t; t, y) = g(z_t; t, \mathbf{B}) = \sum_{\mathcal{B}_i \in \mathbf{B}} \mathcal{L}_r^i(z_t; \mathcal{B}_i) + \mathcal{L}_b^i(z_t; \mathcal{B}_i). \tag{14}$$

At each denoising step, we calculate the gradient of $g(z_t; t, \mathbf{B})$ and update the latent $z_t$:

$$z_t \leftarrow z_t - \eta_g \nabla_{z_t} g(z_t; t, \mathbf{B}). \tag{15}$$

A visual example for the controlling process of cross-attention maps is shown in Appendix D (Figure 7). The proposed R&B iteratively refines the cross-attention maps during the sampling process, and effectively enforces the high responses to concentrate within the corresponding boxes.

## 4 EXPERIMENTS

### 4.1 EXPERIMENTAL SETUP

**Dataset.** We utilize different datasets to validate the effectiveness of our approach quantitatively. First, we compare our model performance against different state-of-the-art methods from the perspective of generation accuracy. We use two benchmarks: HRS (Bakr et al., 2023) and Drawbench (Saharia et al., 2022). The **HRS** dataset is composed of various prompts tagged with name of objects and corresponding labels. We select three tracks to validate the effectiveness of our proposed method: spatial/size/color, the number of prompts for each category are 1002/501/501 respectively. The **Drawbench** dataset comprises 39 prompts, with manually annotated labels assigned to each prompt. To evaluate the spatial accuracy, we select a subset of 20 samples from this benchmark. The bounding box annotations for the above two benchmarks is generated according to the textual prompts with the aid of GPT-4 (Phung et al., 2023). Second, we assess all the methods in terms of their fidelity to the textual input. We select 100 samples from the **MS-COCO** (Lin et al., 2014) dataset and create triplets consisting of image caption, object phrases and bounding boxes. Five images are randomly generated for each samples to measure the consistency with image caption.

**Metrics.** In order to evaluate the generative accuracy, we adopt the method used in HRS (Bakr et al., 2023). This method leverages an object-detection model to detect objects within the synthetic images. Accordingly, one image is considered as a correct prediction when all detected objects are correct and their spatial relationships, sizes or color align with the corresponding phrases in the prompt. For the alignment with layout conditions, we report mean IoU score between the bounding boxes predicted by detection model and the ground-truth. For the fidelity to the textual input, we utilize CLIP-Score to measure the distance between input textual features and generated image features.

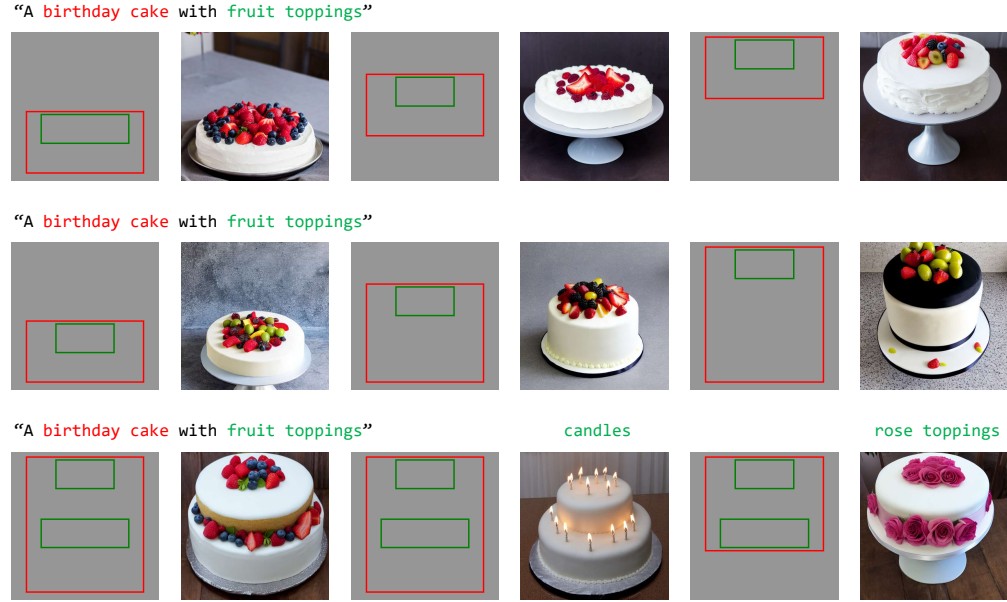

Figure 3: **Illustration of visual variations.** We vary the textual prompt and layout constraints to generate different images. The size and localization of the bounding boxes are changed for spatial control. The instances of the text prompt are alternated for semantic editing. More visual variations for different text prompts can be found in Appendix F (Figure 14).

## 4.2 EXPERIMENTAL RESULTS

**Visual variations.** To validate the robustness of the proposed method, we generate different images by varying the textual input and bounding box constraints. Visual results are shown in Figure 3, we choose "A birthday cake with fruit toppings" as the basic text prompt. In the top row, the localization of the birthday cake changes from the bottom to the top of the image according to the box instructions. In the middle row, the cake grows taller from left to right, and the localization of the fruit toppings changes accordingly. These show that our synthetic images highly align with the box information across a range of size and localization variations. In the bottom row, we adopt multiple changes with prompt alternation and box adjustment, while R&B presents delicate results respectively. This demonstrates that our method simulates the layout control from different aspects robustly, maintaining high generation fidelity and accuracy.

**Quantitative comparison.** We validate the generative accuracy of our method and existing state-of-the-art zero-shot grounded T2I approaches (details can be found in Appendix B). Comparison results are shown in Table 1. For the spatial category, our method surpasses the best methods by 5.69% on HRS and 11.50% on DrawBench. For the size and color categories, our proposed method also beats the baselines by a large margin of 9.77% and 8.5%. The proposed region-aware loss guides the generative process to follow the layout more faithfully, which is the key to improve performance on spatial and size category. Our proposed boundary-aware loss enables the generation of more objects according to layouts, which boosts performance on hard cases with 3 or 4 objects. Quantitative comparisons of the alignment with layout constraints and textual input are in the Appendix F (Table 5).

| Methods | | Stable Diffusion | BoxDiff | Layout-guidance | Attention-refocusing | R&B (Ours) |
|---|---|---|---|---|---|---|
| HRS | Spatial | 8.48 | 16.31 | 16.47 | 24.45 | **30.14** |
| | Size | 9.18 | 11.02 | 12.38 | 16.97 | **26.74** |
| | Color | 12.61 | 13.23 | 14.39 | 23.54 | **32.04** |
| DrawBench | Spatial | 12.50 | 30.00 | 36.50 | 43.50 | **55.00** |

Table 1: **Quantitative comparisons with competing methods.** The evaluation accuracy (%) is reported. Best results are **bold**.

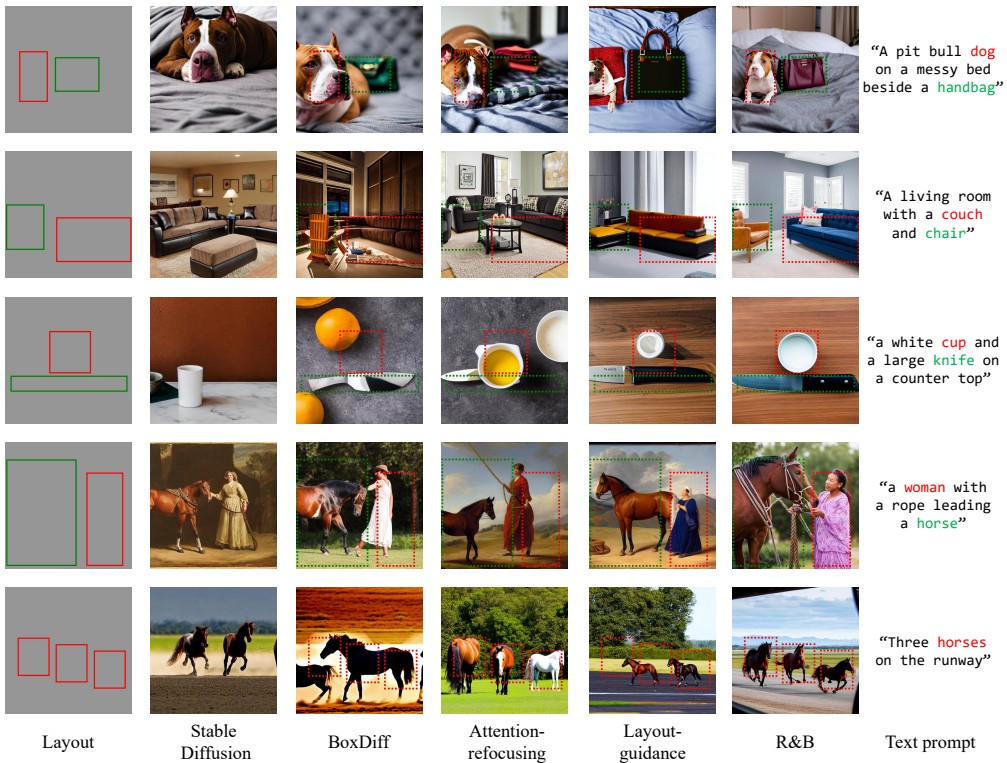

Figure 4: **Visual comparisons with different baselines.** We compare our R&B with Stable Diffusion and several zero-shot grounded T2I baselines. Bounding boxes related to different objects are annotated on the images with dashed boxes. Zoom in for better visualization.

**Visual comparisons.** In Figure 4 we illustrate a visual comparison between our method and several competing baselines. We observe that without additional layout constraints, the vanilla Stable Diffusion suffers from missing objects and object counting error, therefore the generated images fail to convey the given layout information. Existing methods achieve zero-shot grounded T2I generation via diffusion guidance, and present competitive results that better reflects the object layouts. Yet the critical problems of T2I methods mentioned above still exist. For example, BoxDiff fails to generate the cup (3rd row), Attention-refocusing fails to generate the handbag (1st row) and chair (2nd row), Layout-guidance fails to generate the chair (2nd row), and both BoxDiff and Layout-guidance generate horses with incorrect number, see the last row. Furthermore, many synthetic images do not comply well with the layout constraints. **In comparison**, our proposed R&B (1) handles the semantic

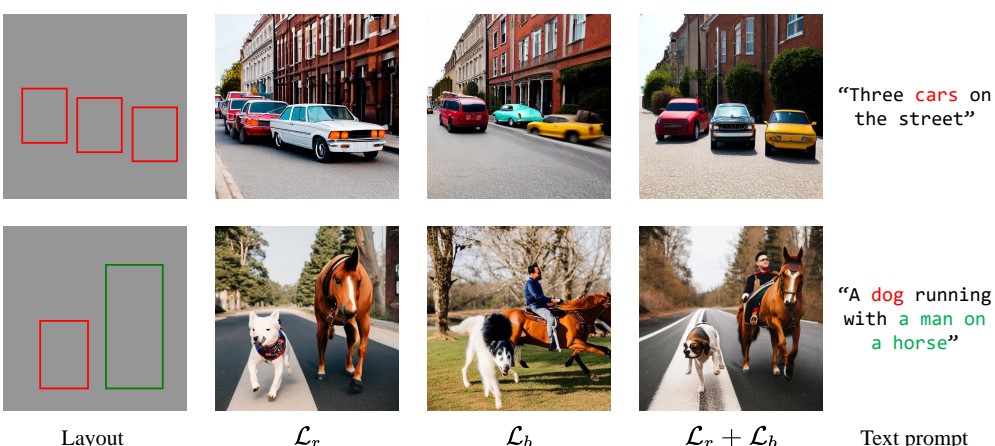

Figure 5: **Ablation study for the effect of different losses.** We illustrate the visual impact of $\mathcal{L}_r$ and $\mathcal{L}_b$ respectively, and also show the combined results.

deficiency problems well, (2) produces images that better align with the bounding boxes, (3) better reflects the semantics of the given text prompts (*e.g.*, rope in the woman's hands for leading a horse). The effectiveness of our R&B method can be attributed to the reliable guidance provided by the region-aware loss and the boundary-aware loss. The region-aware loss accurately captures the discrepancy between the generated layout and the ground-truth boxes, while the boundary-aware loss calibrates the generative process by encouraging the expression of different objects in their corresponding regions. We notice that although Attention-refocusing yields high quantitative performance on HRS and Drawbench benchmarks, the generated images do not align well with the box constraints. The reason is that the above two benchmarks only evaluate the model's generation accuracy in terms of coarse-grained aspects. Images that do not align with the bounding box can still be considered as correct in the evaluation. Additional generative results under different random seeds for competing methods can be found in Appendix F (Figure 9).

### 4.3 ABLATION STUDY

**Impact of different loss components.** In Figure 5, we present visual results for comprehension of our proposed region-aware loss $\mathcal{L}_r$ and boundary-aware loss $\mathcal{L}_b$. We observe that $\mathcal{L}_r$ (the second column) contributes to good spatial accuracy, yet does not handle some difficult semantics well. For example, the wrong number of cars, and the missing man on the horse. On the other hand, with $\mathcal{L}_b$ alone, the generated images (the third column) better adhere to the text semantics, but do not align well with the ground-truth bounding boxes. The fourth column shows visual result of the combination of the two losses. We witness that they complement each other well, and present boosted generative results with higher spatial and semantic accuracy. As for the quantitative analysis of the two losses, please refer to Table 4 in Appendix E.

**Impact of the guidance ratio.** We analyze the impact of the guidance ratio $\eta_g$ in Eq. (15) for the energy function $g$ from the quantitative aspect. Numerical results on MS-COCO are shown in Table 2. We scale $\eta_g$ from 20 to 300, and report the mean IoU and T2I similarity to measure the alignment with layout and textual input respectively. We utilize UniDet (Zhou et al., 2022) to detect objects in the synthesized images, and calculate the mean IoU score between the predicted object bounding boxes and their corresponding ground-truth boxes. The upper bound of the mean IoU is 0.875, for some phrases in the textual prompts do not belong to the test classes of UniDet. As for the text-to-image similarity, we calculate the CLIP-Score of the image features and corresponding text prompt features. We observe the fact that as $\eta_g$ grows, the two scores are improved at the beginning, and then experience a rapid decline. The reason is that excessively strong constraints greatly impair the generative fidelity, leading to a deterioration in the evaluation results. Practically, we choose $\eta_g$ as 70 for all the experiments, in order to balance the generative accuracy and fidelity. Due to space constraints, we discuss the additional ablation results detailly in Appendix E.

|  | 20 | 50 | 60 | 70 | 80 | 100 | 150 | 300 |
|---|---|---|---|---|---|---|---|---|
| mIoU ($\uparrow$) | 0.5075 | 0.5303 | 0.5239 | 0.5533 | 0.5460 | 0.5658 | 0.5626 | 0.4818 |
| T2I-Sim ($\uparrow$) | 0.3146 | 0.3141 | 0.3186 | 0.3218 | 0.3138 | 0.3158 | 0.3136 | 0.3007 |

Table 2: **Quantitative analysis of the guidance ratio.** We utilize mIoU to measure the alignment with box instructions, and CLIP-Score to measure the correspondence between images and texts.

## 5 CONCLUSION

In this paper, we propose R&B, a novel attention guidance approach for diffusion based zero-shot text-to-image generation. We point out two prevalent issues associated with the current approaches: (1) the misalignment between generated images and layout information, and (2) the semantic issues inherited from the base T2I models. Accordingly, we propose a region and boundary aware guidance approach to mitigate the above problems. Experimental results show that our proposed method outperforms the existing baselines by a large margin both quantitatively and qualitatively, and verify its robustness to various layout and textual conditions. Future work will focus on extensions of zero-shot grounded generation in different scenarios (*e.g.*, 3D, video).

## 6 ACKNOWLEDGEMENT

This work was supported in part by National Natural Science Foundation of China: 62322211, 61931008, 62236008, 62336008, U21B2038, the Key R&D Plan Project of Zhejiang Province (No. 2024C01023), and Youth Innovation Promotion Association of Chinese Academy of Sciences under Grant 2020108.

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

# A  IMPLEMENTATION DETAILS

We utilize the Stable Diffusion V-1.5 (Rombach et al., 2022) as our base generative model. We adopt the DDIM scheduler (Song et al., 2020a) with 50 denoising steps. The ratio of classifier-free guidance is set as 7.5. Previous works (Balaji et al., 2022; Hertz et al., 2022; Choi et al., 2022) show that T2I diffusion models tend to form the layout of each object in early stages and refine the generative details (color, textures) in later stages. Thus we only perform layout guidance at the first 10 steps. The $\lambda$ for dynamic threshold in Eq. (6) is set as 0.4, the ratios $\lambda_s$ and $\lambda_a$ in Eq. (11) are set as 1.5 and 1.0 respectively.

# B  DETAILS OF COMPETING METHODS AND DISCUSSIONS

**Layout-guidance** (Chen et al., 2023) is an attention control method for zero-shot grounded T2I generation. It designs an energy function according to the box constraints:

$$E(A^{(\gamma)}, B, i) = (1 - \frac{\sum_{u \in B} A_{ui}^{(\gamma)}}{\sum_u A_{ui}^{(\gamma)}})^2, \tag{16}$$

where $A_{ui}^{(\gamma)}$ determines the attention value at location $u$ corresponded to the $i^{\text{th}}$ token for the layer $\gamma$. This energy function enforces high activation within the masked regions. At each time step, the gradient of $E$ is computed to update the noisy latent through backpropagation. Although adopting energy function in Eq. (16) for layout guidance demonstrates effectiveness in many situations, there still remains some critical issues. (1) Optimizing Eq. (16) aims at concentrating all attention scores within the bounding box, which may lead to some tricky solutions (Xie et al., 2023). For example, in Figure 4 ($3^{\text{rd}}$ row, $5^{\text{th}}$ column), the generated cup is much smaller than the corresponding box. And the number of generated horses is incorrect ($5^{\text{th}}$ row, $5^{\text{th}}$ column), because most of the class-related areas are within the boxes, the designed energy function is fooled and does not provide accurate latent optimization directions. (2) This energy function is not sensitive to the scale of the activations and may not effectively address some semantic deficiencies issues (missing objects, attribute binding). Thus, even though Layout-guidance presents some delicate visual results in Figure 4, it does not perform well on the HRS and Drawbench benchmarks.

**BoxDiff** (Xie et al., 2023) proposes three constraints for training-free grounded T2I image generation: Inner-Box, Outer-Box, and Corner constraints. The Inner-Box constraint selects top-k elements of cross-attention map within the bounding box and maximize their values. The Outer-Box constraint selects top-k elements out of the box and minimize their values. The Corner constraint projects the attention values alongside the x-axis and y-axis of the bounding boxes, and maximizes the projected values in order to cope with some tricky solutions (*e.g.*, the object becomes much more smaller than the ground-truth box). The key insight of BoxDiff is to impose constraints on fewer elements with higher responses of cross-attention maps, rather than all of the elements. When reproducing BoxDiff, we find that it heavily relies on choice of initial seeds. The main reason is that the top-k selection strategy introduces uncertainty during the denoising process. As a result, the generated outcomes exhibit significant instability. This is shown in Figure 9 and 16, the generative layouts varies greatly across different seeds, picking is required to adopt delicate results. We also observe that only impose constraints on the peak values may results in outcomes that only discriminative regions of objects locate within the box, see Figure 9 (upper half, $1^{\text{st}}$ row), Figure 16 (bottom half, $1^{\text{st}}$ row). Further, visual results (Figure 4, Figure 9 and Figure 16) show that BoxDiff does not effectively cope with the semantic issues.

**Attention-refocusing** (Phung et al., 2023) proposes a self-attention refocusing and cross-attention refocusing (SAR&CAR) approach to guide the diffusion process according to the layout constraints by manipulating both cross-attention maps and self-attention maps. As for the cross-attention guidance, it maximizes the top values of cross-attention maps within bounding boxes in the same way as the previous method (Chefer et al., 2023), and penalizes the top values in other object regions and background respectively to suppress incorrect expression of different objects. As for the self-attention guidance, it minimizes the similarity between the elements of background regions and the inner-box regions to enforce the expression of objects in the correct places. Attention-refocusing is essentially an enhancement of training-based methods (*e.g.*, GLIGEN (Li et al., 2023a)), it borrows inspiration

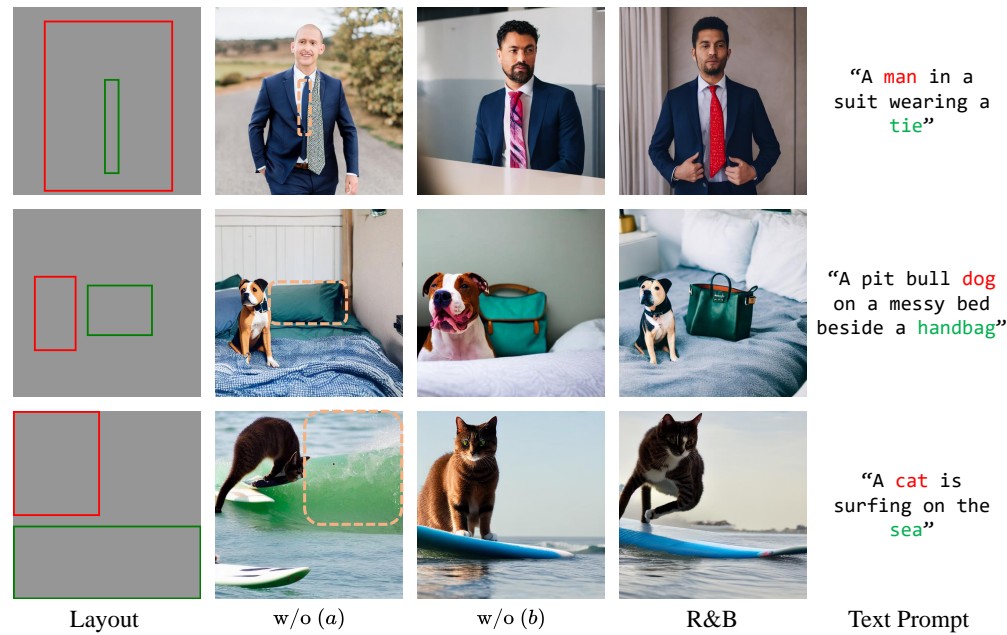

Figure 6: **Visual examples for the effect of (a)** $\mathcal{M}_i^s$ **and (b)** $\mathcal{M}_i^a$**.** We illustrate the two variants of cross-attention maps respectively, and also show the combined results.

from previous work (Chefer et al., 2023) to optimize the max attention values within and out of the input boxes to cope with the semantic issues in grounded generation. Quantitative results in Table 1 and 6 show that its proposed SAR&CAR boosts the performance of existing T2I methods like Stable Diffusion and GLIGEN. However, when regarded as a training-free grounded generation approach, the generative outcomes lack spatial accuracy. See Figure 4, 9 and 16 for visual examples. Similar to BoxDiff, optimizing the max values of attention maps incorporates randomness into the generative procedure and only ensures the most discriminative regions of objects are within the layout. Meanwhile, the self-attention refocusing may not be effective during the early layout refinement process, for it is not aware of the textual condition, when the generative layout differs significantly from the ground truth, it does not provide accurate guidance for corresponding ojbects.

## C    VARIANTS OF AGGEGATED CROSS-ATTENTION MAPS

We mentioned two variants of $\mathcal{M}_i$ in Section 3: $\mathcal{M}_i^s$ and $\mathcal{M}_i^a$, respectively. Here we formulate these two variants as below:

$$\mathcal{M}_i^s = \texttt{normalize}(\texttt{sigmoid}(s \cdot (\mathcal{M}_i^{\text{norm}} - \tau_i))), \tag{17}$$

$$\mathcal{M}_i^a = \mathcal{M}_i^{\text{norm}} = \frac{\mathcal{M}_i - \min_{h,w}(\mathcal{M}_i)}{\max_{h,w}(\mathcal{M}_i) - \min_{h,w}(\mathcal{M}_i)}. \tag{18}$$

The formulation of $\mathcal{M}_i^s$ is inspired by previous work (Epstein et al., 2023), which adopts soft masks from the cross-attention maps. The $\texttt{sigmoid}(\cdot)$ operation (with sharpness controlled by $s$) helps to eliminate the noisy background that is thresholded by $\tau_i$. In this manner, $\mathcal{M}_i^s$ represents a soft mask that describes the shape and size of the $i^{\text{th}}$ object. Practically, we find that although imposing constraints on $\mathcal{M}_i^s$ helps for fast convergence of layout control, it is somehow not sufficient for grounded T2I generation. This is because the $\texttt{sigmoid}(\cdot)$ function introduces nonlinearity to the guidance process. When the responses of elements deviate from the threshold, the gradients of the elements decrease rapidly. This leads to the insensitivity to the layout guidance in some local regions (e.g., background regions), reflected by redundant textures or catastrophic neglect in the output images. To cope with the above issue, we further utilize $\mathcal{M}_i^a$, which simply rescales the cross-attention maps with linear transforms, as a complement for $\mathcal{M}_i^s$. $\mathcal{M}_i^a$ preserves the quantitative relationships between different activation regions of the original attention map, thereby better reflecting the appearance

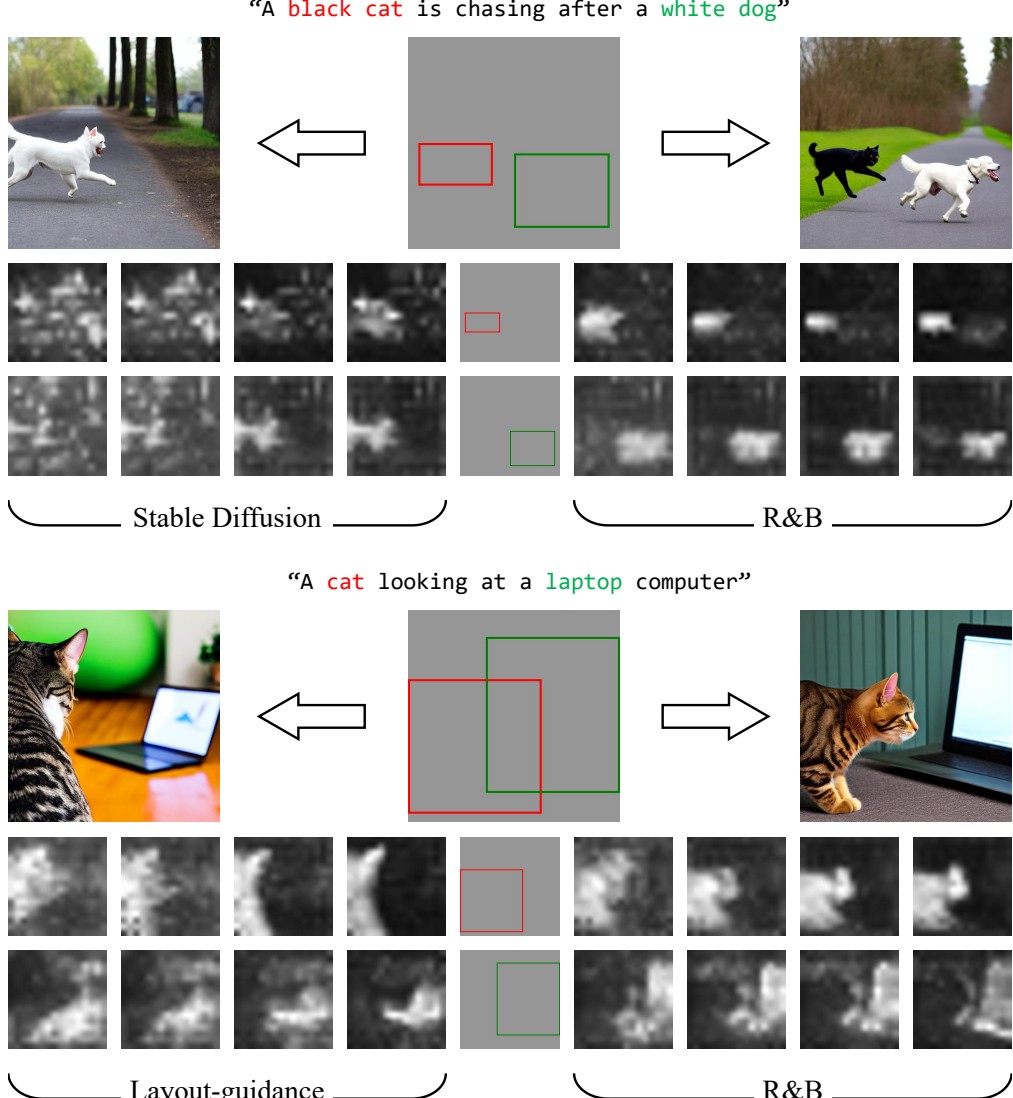

Figure 7: **Visualization of cross attention maps** at earlier diffusion steps. We show visual comparison between our R&B and (1) Stable Diffusion (Rombach et al., 2022), (2) Layout-guidance (Chen et al., 2023). Both Layout-guidance and our method perform layout guidance at the first 10 steps.

information associated with the attention map. For better comprehension, we illustrate a visual comparison in Figure 6, (a) and (b) represent using $\mathcal{M}_i^a$ and $\mathcal{M}_i^s$ for layout guidance respectively. We find that only optimizing $\mathcal{M}_i^s$ (2nd column) leads to weird visual effects in the synthetic images annotated by dashed boxes, for example, the redundant tie (1st row) and sea waves (last row), and the absence of the handbag on the bed (2nd row). By comparison, optimizing $\mathcal{M}_i^a$ (3rd column) overcomes the above issues, yet does not guarantee alignment with the bounding box constraints (*e.g.*, the wrong size of man, dog, and handbag in the first two rows). When optimizing the two variants jointly during guidance process, R&B presents images with high (1) consistency with the layout conditions, and (2) fidelity to the textual semantics.

## D    VISUALIZATION OF CROSS-ATTENTION MAPS

The first few denoising steps of diffusion models decide the approximate spatial layout of the generated image, while cross attention maps are an intuitive way to describe it. To illustrate how our method controls the generative layout through attention maps, we provide visualization of cross

attention maps within 10 denoising steps in Figure 7. The attention map is gradually refined from left to right as the number of denoising steps increases. The upper half compares our method to Stable Diffusion (Rombach et al., 2022) and the lower half compares our method to Layout-guidance (Chen et al., 2023). Stable diffusion lacks guidance of spatial information, therefore both objects are possible to compete and activate on the same region, which leads to missing objects and attribute misbinding in the generated image. Layout-guidance restricts high activation regions within the corresponding bounding boxes, but fails to align the shape and size of objects with the box constraints. By comparison, under the guidance of our region and boundary aware loss, the generative model precisely localizes objects according to their layouts in the early sampling process. With the well initialized generative layout, the T2I diffusion model produces images that align with the layout conditions precisely, and possess high generative fidelity.

## E  MORE ABLATIONS AND ANALYSIS

**Dynamic thresholding.** Thresholding class-related activation maps (Vinogradova et al., 2020; Huang et al., 2018) for precise foreground region extraction is a challenging task, the reason is that only weakly supervised information is available. Previous work (Epstein et al., 2023) adopts a fixed threshold to obtain the size and shape of objects from the diffusion cross-attention maps for image editing. Yet we find this not effective for layout generation, for the reason that the distribution of cross-attention maps changes during guidance process. To address this, we utilize a dynamic thresholding approach to highlight the foreground regions according to the mean activation values within and outside the bounding box, formulated in Eq. (6). We show visual results in Figure 8. We find that utilizing fixed threshold faces several issues: (1) it struggles to accurately generate objects, which the original Stable Diffusion fails to synthesize, in the corresponding regions (*e.g.*, the branch in the 1st row and the hello kitty in the 2nd row). (2) it does not solve the improper attribute binding problem, like the confusion of horse and car in the bottom row.

**Discrete sampling.** Inspired by previous works on deep model quantization (Jang et al., 2016; Sohn et al., 2015), we differentiably binarize the cross-attention maps via a Straight-Through Estimator (STE), the formulation is shown in Eq. (8) and Eq. (9). This operation transforms the consecutive

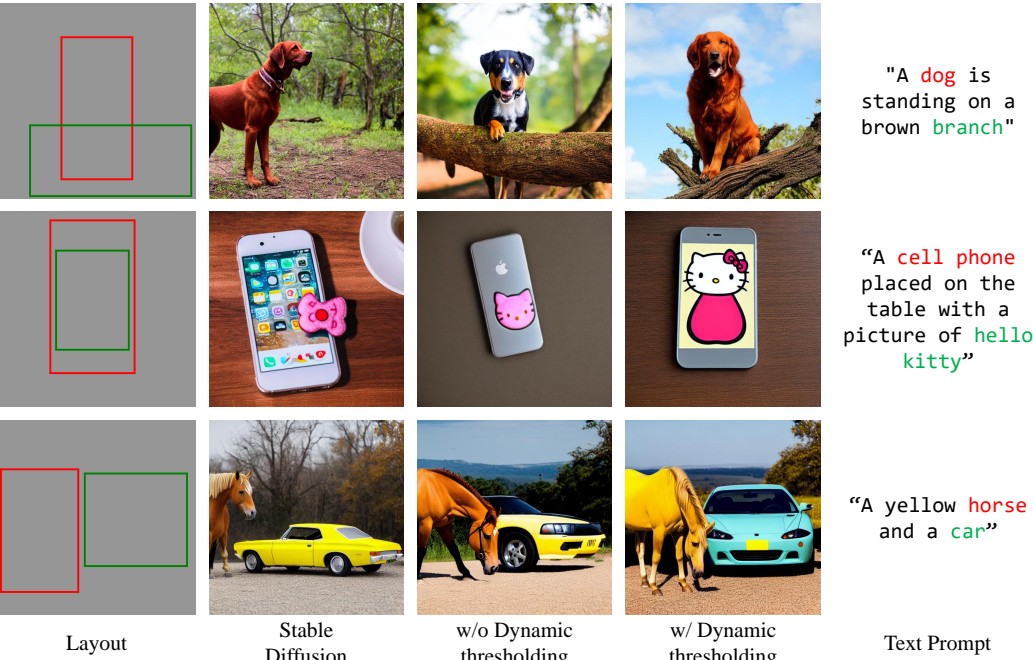

Figure 8: **Visualization of the impact of dynamic thresholding.** We show visual results of the original Stable Diffusion (2nd column), binarizing the attention maps with threshold 0.5 (3rd column), and binarizing the attention maps with a dynamic threshold (4th column), respectively.

|  | w/o Discrete sampling | w/ Discrete sampling |
|---|---|---|
| mIoU ($\uparrow$) | 0.5106 | 0.5533 |
| T2I-Sim ($\uparrow$) | 0.3142 | 0.3218 |

Table 3: **Quantitative ablation of discrete sampling.** We conduct experiments on the subset of MS-COCO. All the hyper-parameters are fixed for fair comparison.

attention maps into differentiable hard masks, and we design a discrete energy function in Eq. (11) to directly measure the divergence between the predicted bounding boxes $\hat{\mathcal{B}}_i$ with the ground-truth $\mathcal{B}_i$.

We show the impact of discrete sampling approach from the quantitative perspective, and illustrate the results in Table 3. We alternate the discrete maps in Eq. (11) with $\mathcal{M}_i^s$ and $\mathcal{M}_i^a$ to ablate the effect of discrete sampling. We observe that with the aid of discrete sampling, the generated images better align with the box instructions and adhere to the textual input semantically.

**Quantitative analysis of the proposed two losses.** Here we analyze the impact of our proposed two loss functions: region-aware loss $\mathcal{L}_r$ and boundary-aware loss $\mathcal{L}_b$ quantitatively. The generative accuracy (%) on HRS and Drawbench is reported in Table 4. We observe that both $\mathcal{L}_r$ and $\mathcal{L}_b$ greatly boost the performance of the original Stable Diffusion because they incorporate the layout information into the generative process. Generally, using $\mathcal{L}_r$ for guidance yields higher accuracy than using $\mathcal{L}_b$, especially in terms of the size metric on HRS. This is because $\mathcal{L}_r$ models the discrepancy between cross-attention maps and bounding-box constraints more precisely, and the shape and localization of generated objects better align with the layout instructions. Moreover, we find that the performance of the diffusion model gets further improved when combining the above two losses together. This demonstrates that these two losses cooperate with each other well during the guidance process.

| Methods | | Stable Diffusion | w/o $\mathcal{L}_r$ | w/o $\mathcal{L}_b$ | R&B |
|---|---|---|---|---|---|
| | Spatial | 8.48 | 20.01 | 25.99 | 30.14 |
| HRS | Size | 9.18 | 15.05 | 23.26 | 26.74 |
| | Color | 12.61 | 23.22 | 27.32 | 32.04 |
| DrawBench | Spatial | 12.50 | 42.50 | 47.50 | 55.00 |

Table 4: **Quantitative results for different constraints.** We evaluate two variants, where $\mathcal{L}_r$ and $\mathcal{L}_b$ are removed from our full method. The generative accuracy (%) on HRS and DrawBench is reported.

# F   ADDITIONAL RESULTS

**Qualitative comparisons.** In Figure 9 we provide qualitative comparisons among different baselines and our proposed R&B. Layout-guidance suffers from severe attribute misbinding problem, while BoxDiff and Attention-Refocusing suffer from missing object. These issues are inhereted from Stable Diffusion, which reflects that guidance from these methods is insufficient for the model to understand the semantics of each object. Moreover, all baselines fail to generate objects that conform to the bounding boxes accurately. Our proposed R&B mitigates the semantic issues by incentivizing object activation in a region and boundary aware manner. It further greatly improves the alignment between generated images and layout conditions through explicitly minimizing the difference between MBRs and ground-truth bounding boxes. As a result, R&B generally produces images that better conform to user input both semantically and spatially. Additionally, we find that R&B consistently produces high-quality images regardless of seed selection, while other baselines rely on cherry picking from a pile of generated images to satisfy user input.

**Quantitative comparisons.** We evaluate the adherence to the box constraints and textual input of generated images for different diffusion-based methods. The experiments are conducted on

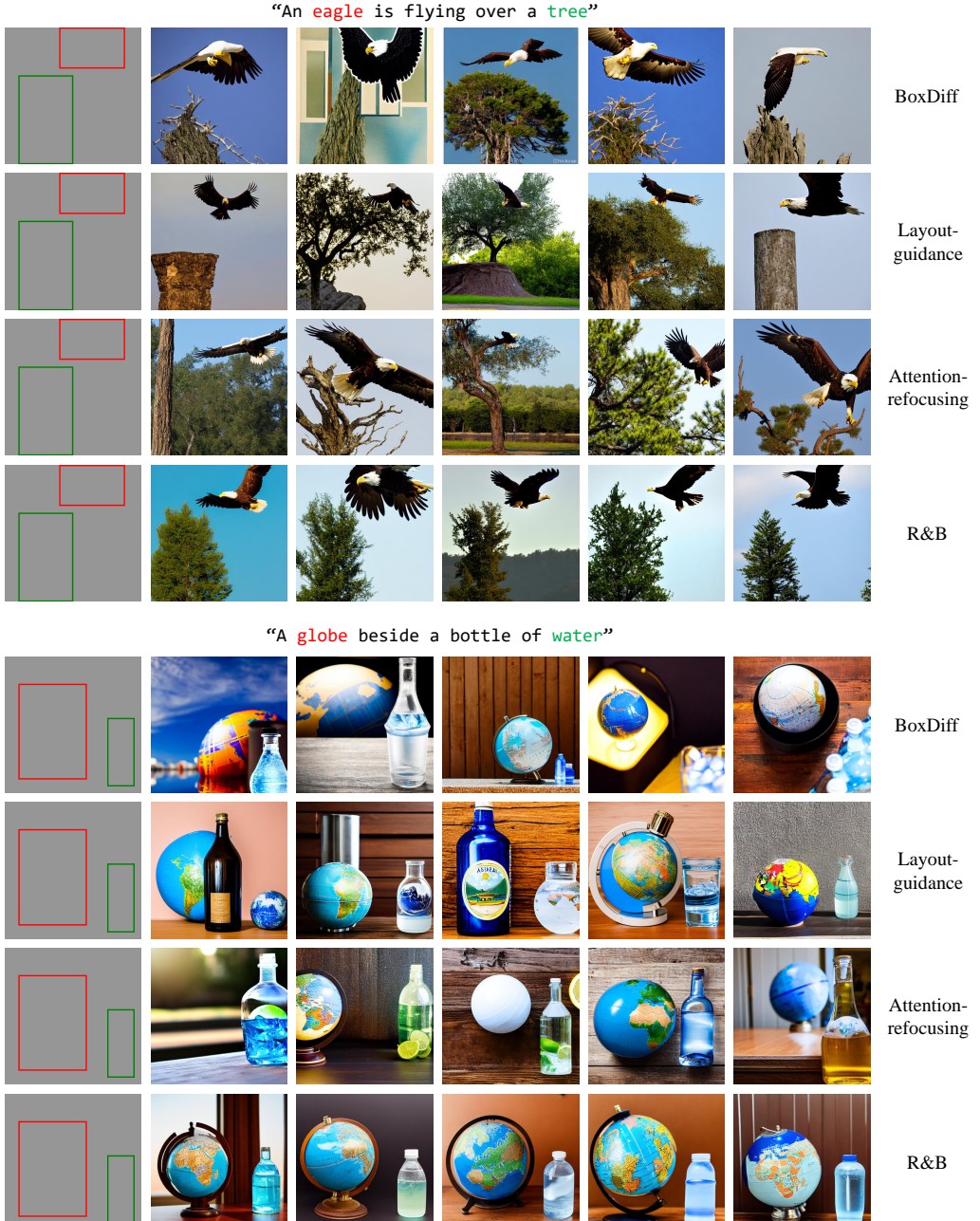

Figure 9: Visual comparisons among Layout-guidance (Chen et al., 2023), BoxDiff (Xie et al., 2023), Attention-Refocusing (Phung et al., 2023) and the proposed R&B.

| Methods | Stable Diffusion | BoxDiff | Attention-refocusing | Layout-guidance | R&B (Ours) |
|---|---|---|---|---|---|
| mIoU (↑) | 0.2700 | 0.3753 | 0.3960 | 0.4460 | **0.5533** |
| T2I-Sim (↑) | 0.2920 | 0.3003 | 0.3023 | 0.3108 | **0.3218** |

Table 5: **Quantitative comparisons with baselines.** We calculate the mean IoU and CLIP similarity for different methods on the selected subset of MS-COCO. For each sample, we randomly generate five images for evaluation. Best results are **bold**.

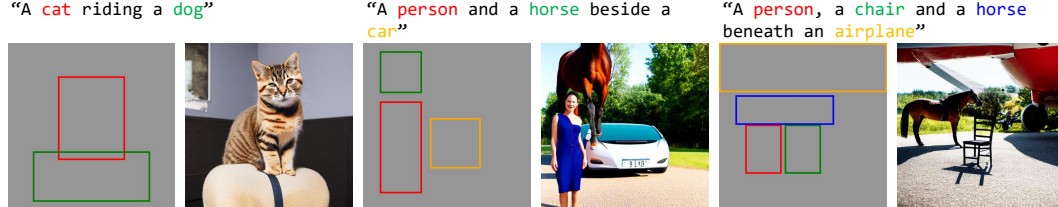

Figure 10: Failure cases of our proposed method.

the manually collected subset of MS-COCO with 100 samples. We observe that the SOTA zero-shot grounded T2I methods BoxDiff (Xie et al., 2023), Layout-guidance (Chen et al., 2023), and Attention-refocusing (Phung et al., 2023) greatly improve the mean IoU of generated images because they effectively incorporate the layout information into the diffusion process. Meanwhile, the T2I similarity of Stable Diffusion is also boosted due to the spatial alignment. By comparison, our proposed R&B surpasses the existing methods by a large margin from the two aspects of fidelity to the text prompt and alignment with the layout condition, which agrees with the visual results shown in Figure 4. Quantitative results further validate that our method helps the model to generate images with better spatial accuracy, and simultaneously convey the essential semantics of the text prompts.

**More qualitative results.** Figure 11, 12, and 13 illustrate synthetic results of the proposed R&B under different random seeds. We observe obvious textural level variations across different random seeds, which shows great diversity of the synthetic results. Yet the objects in the generated images consistently align with the layout conditions. In Figure 14, given a fixed bounding box condition, we vary the key phrases of the original prompt "A castle stands across the lake under the blue sky" to adopt different generative results. Our proposed R&B produces images with (1) high fidelity to the various prompts, and (2) adherence to the layout instructions.

## G    LIMITATIONS AND DISCUSSION

Figure 10 shows three possible failure cases of our proposed method. Firstly, if the prompt is too uncommon, e.g. Out-Of-Domain, our method cannot generate accurate objects according to the given layout. Secondly, unreasonable layouts also worsen the generation, leading to unrealistic objects. This is because the pretrained base model (Stable Diffusion) does not cover these extreme scenes. Thirdly, prompts with too many objects (four and more) are beyond the capacity of model and missing objects are observed.

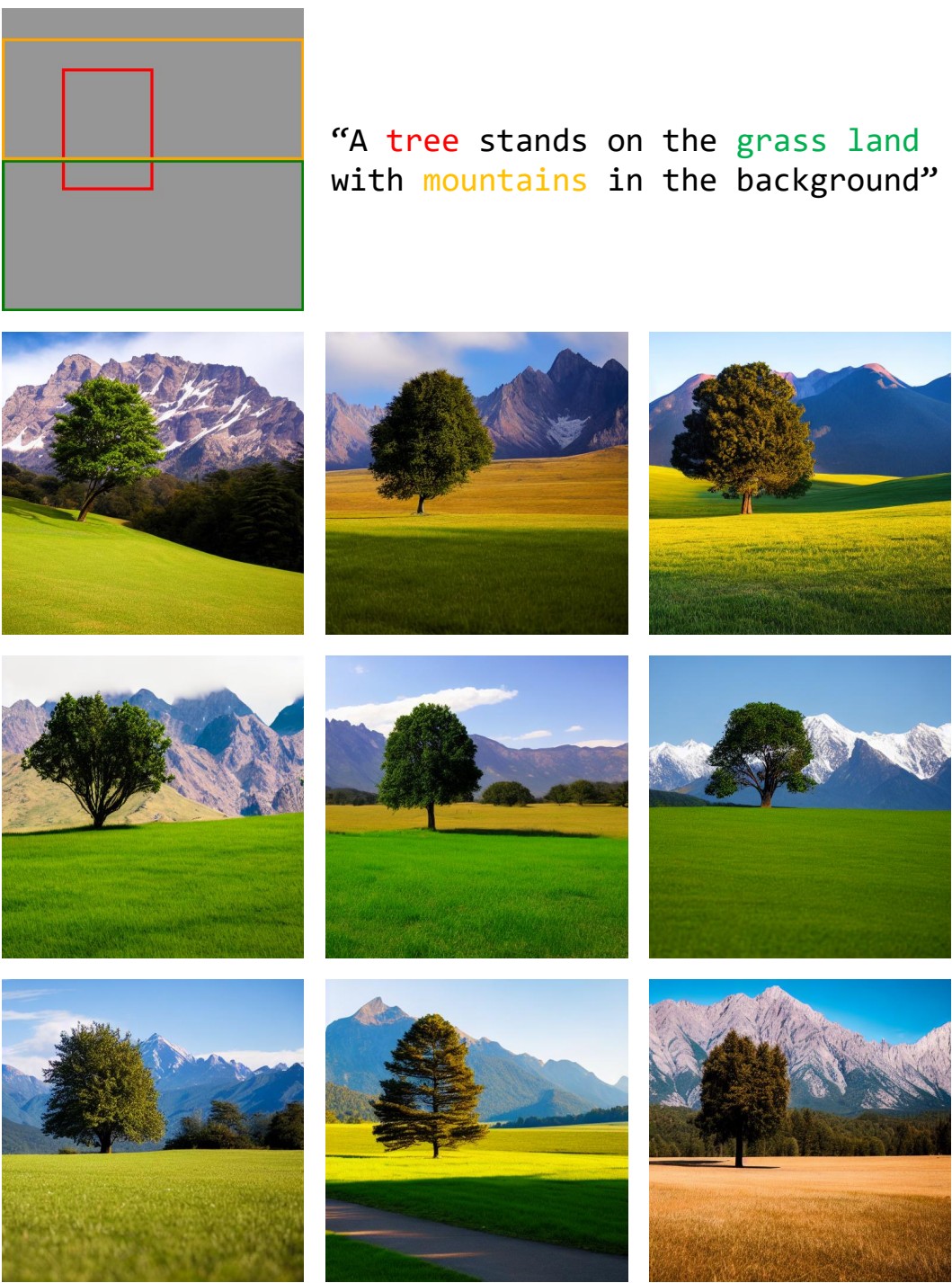

Figure 11: Synthetic trees, grasslands, and mountains with the same spatial conditions.

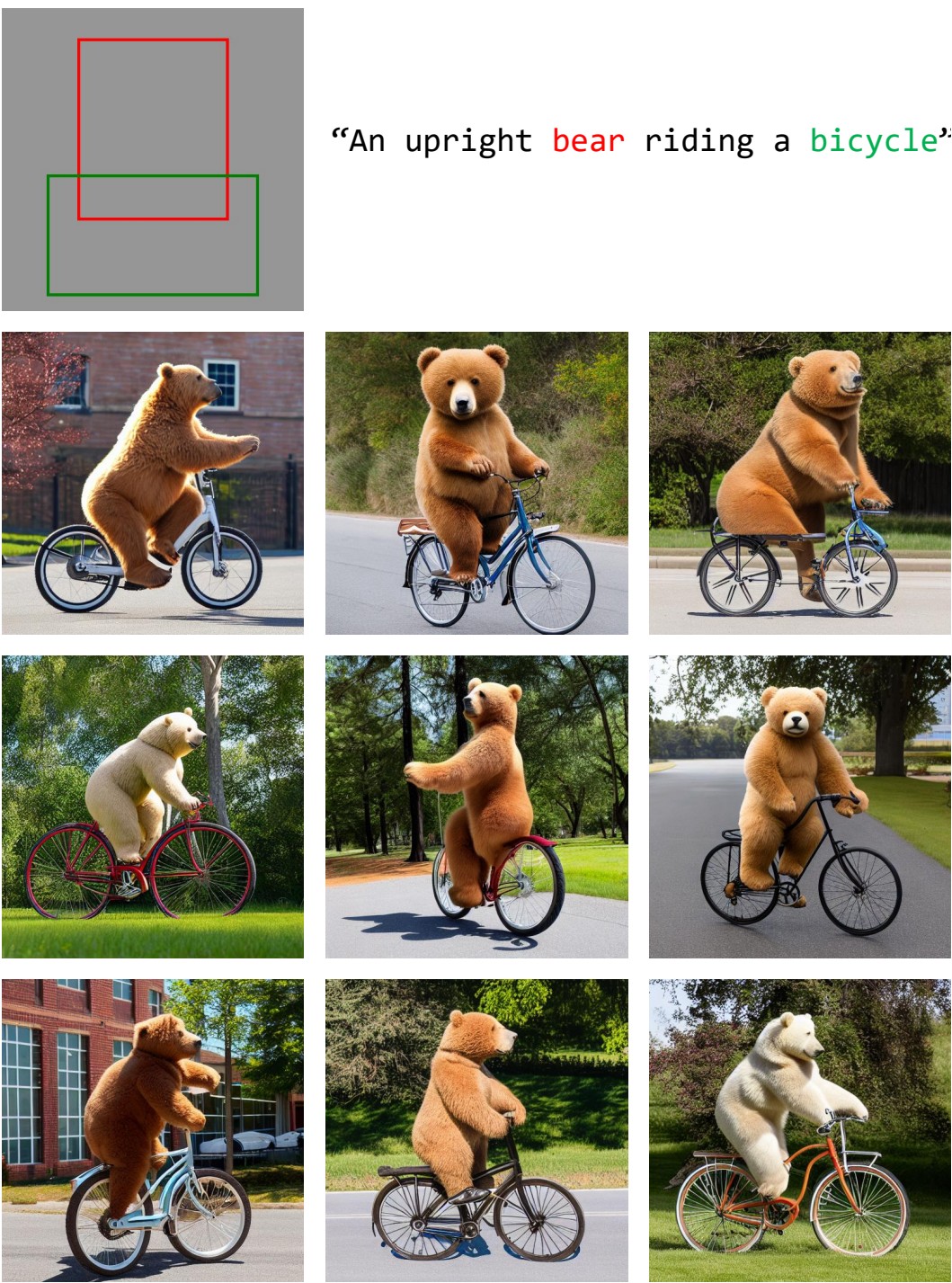

Figure 12: Synthetic bears riding bicycles with the same spatial conditions.

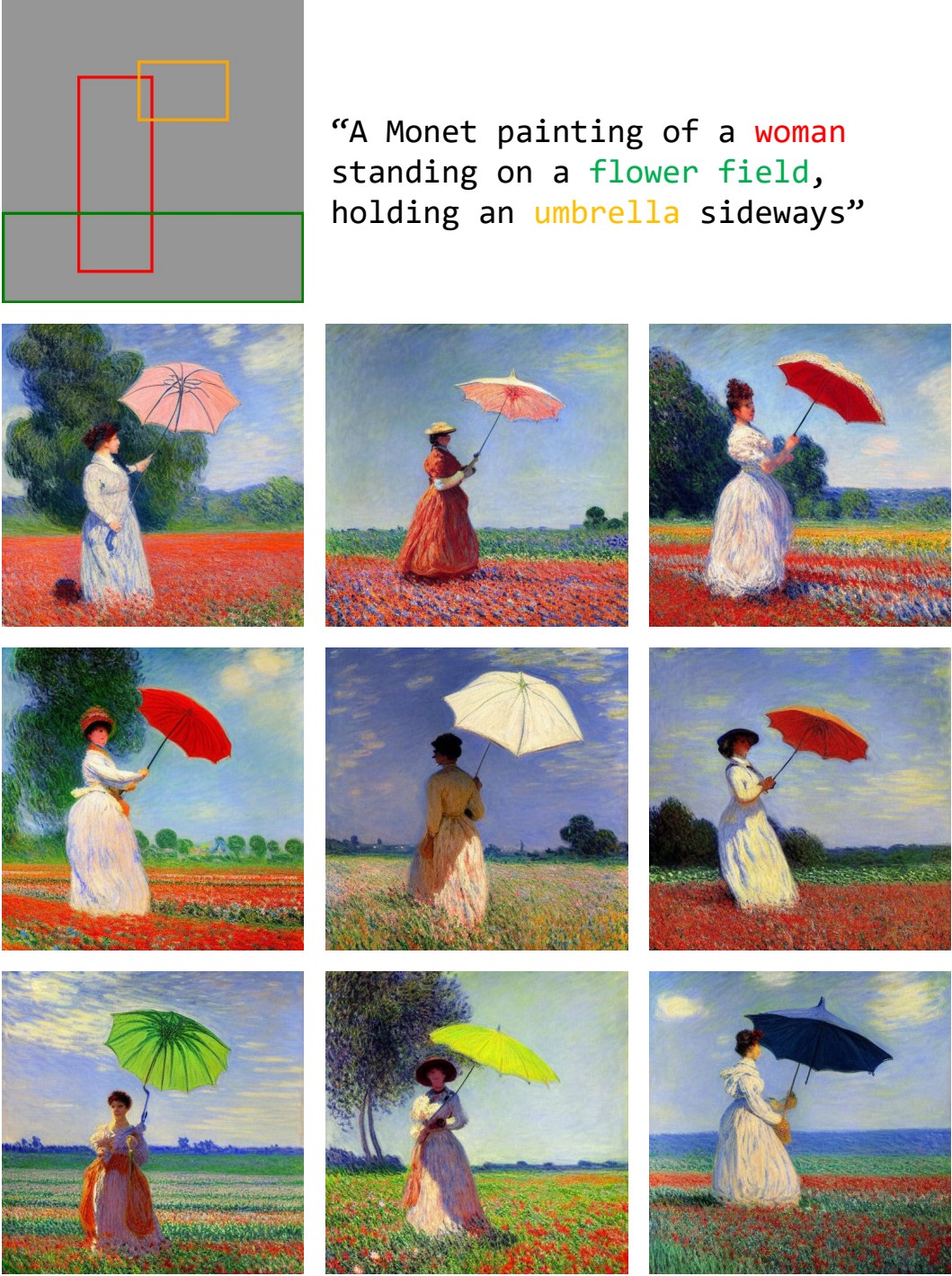

Figure 13: Synthetic women holding umbrellas on flower fields in the style of Monet with the same spatial conditions.

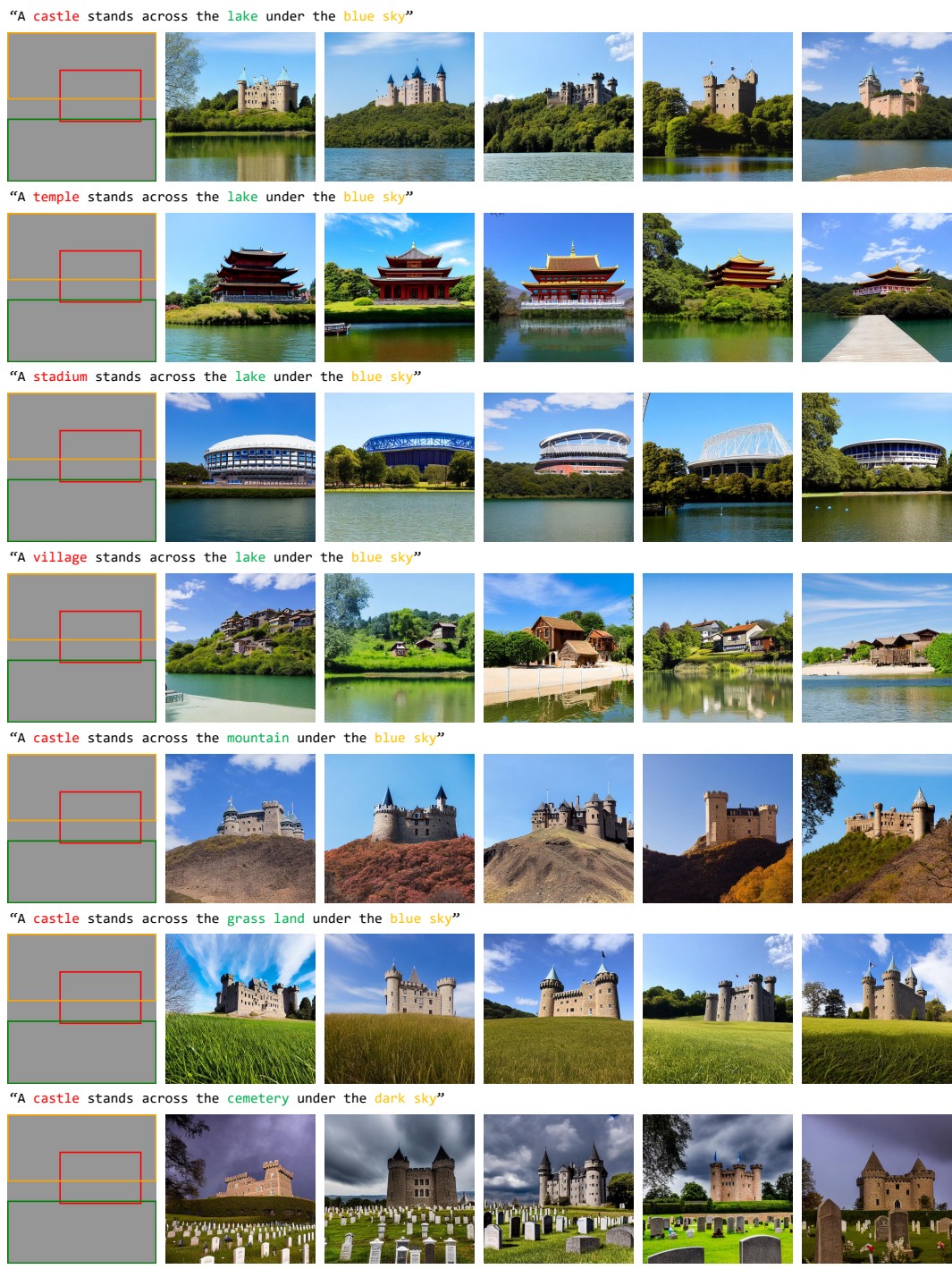

Figure 14: Synthetic different buildings on different backgrounds.

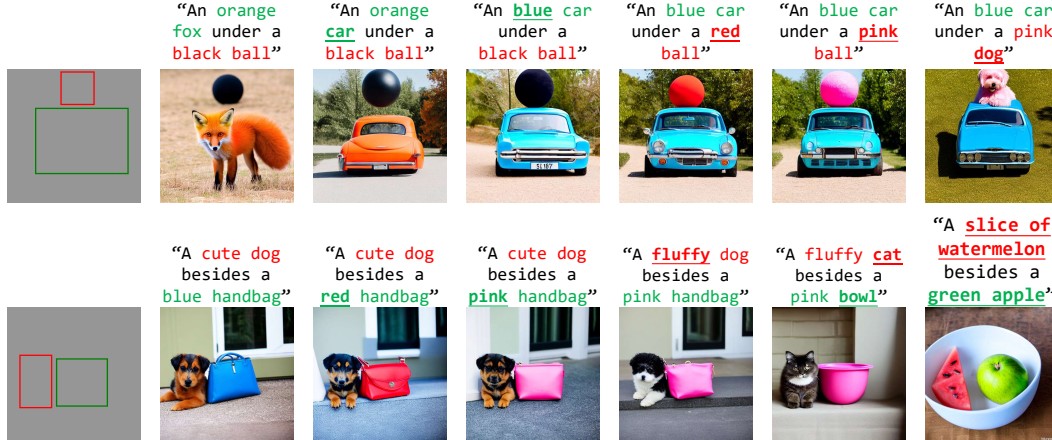

Figure 15: Illustration of the effect of R&B for attribute binding in T2I models. We vary the attributes and nouns in the textual prompts to generate different images. Textual changes are **emphasized**.

## H    VARIATION OF ATTRIBUTES AND NOUNS

In Figure 15 we illustrate how the proposed R&B copes with the attribute binding issues. Given consistent grounding input, we vary the nouns or attributes to adopt different synthetic results. Vanilla Stable Diffusion struggles to allocate multiple attributes with respect to different objects properly, which results in semantic errors. Nevertheless, we witness that R&B helps to bind the nouns and their corresponding attributes across different variations.

## I    COMPARISON WITH GLIGEN

In this section, we first compare our R&B with the training-based grounded T2I generation approach GLIGEN (Li et al., 2023a) from the quantitative aspect. We then illustrate some visual comparisons for better comprehension.

**GLIGEN** searches for embedding the spatial instructions into a pretrained T2I diffusion models (*e.g.*, Stable Diffusion). It injects auxiliary gated self-attention layer at each transformer block to absorb new grounding input. Training Stable Diffusion model on COCO detection datasets requires for 500k iteration on 16 V100 GPUs.

**Quantitative.**   We quantitatively compare our method with GLIGEN on HRS and Drawbench. Numerical results are shown in Table 6. We find that although well-trained on annotated data, GLIGEN does not perform well on the color setting, which implies that it does not effectively cope with the attribute binding issues and some out-of-domain textual prompts (*e.g.*, "a red dog and an orange horse"). As for the size and spatial categories, GLIGEN yields high accuracy because the generated images align well with the size and location of the grounding input. Notably, when

| Methods | | SAR&CAR | R&B | GLIGEN | GLIGEN+SAR&CAR | GLIGEN+R&B |
|---|---|---|---|---|---|---|
| **HRS** | **Spatial** | 24.45 | 30.14 | 45.71 | 54.19 | **56.87** |
| | **Size** | 16.97 | 26.74 | 36.13 | 39.72 | **42.69** |
| | **Color** | 23.54 | 32.04 | 17.84 | 29.46 | **35.72** |
| **DrawBench** | **Spatial** | 43.50 | 55.00 | 48.00 | 64.00 | **67.50** |

Table 6: **Quantitative comparisons with GLIGEN.** The evaluation accuracy (%) is reported. Best results are **bold**. Second best results are underlined.

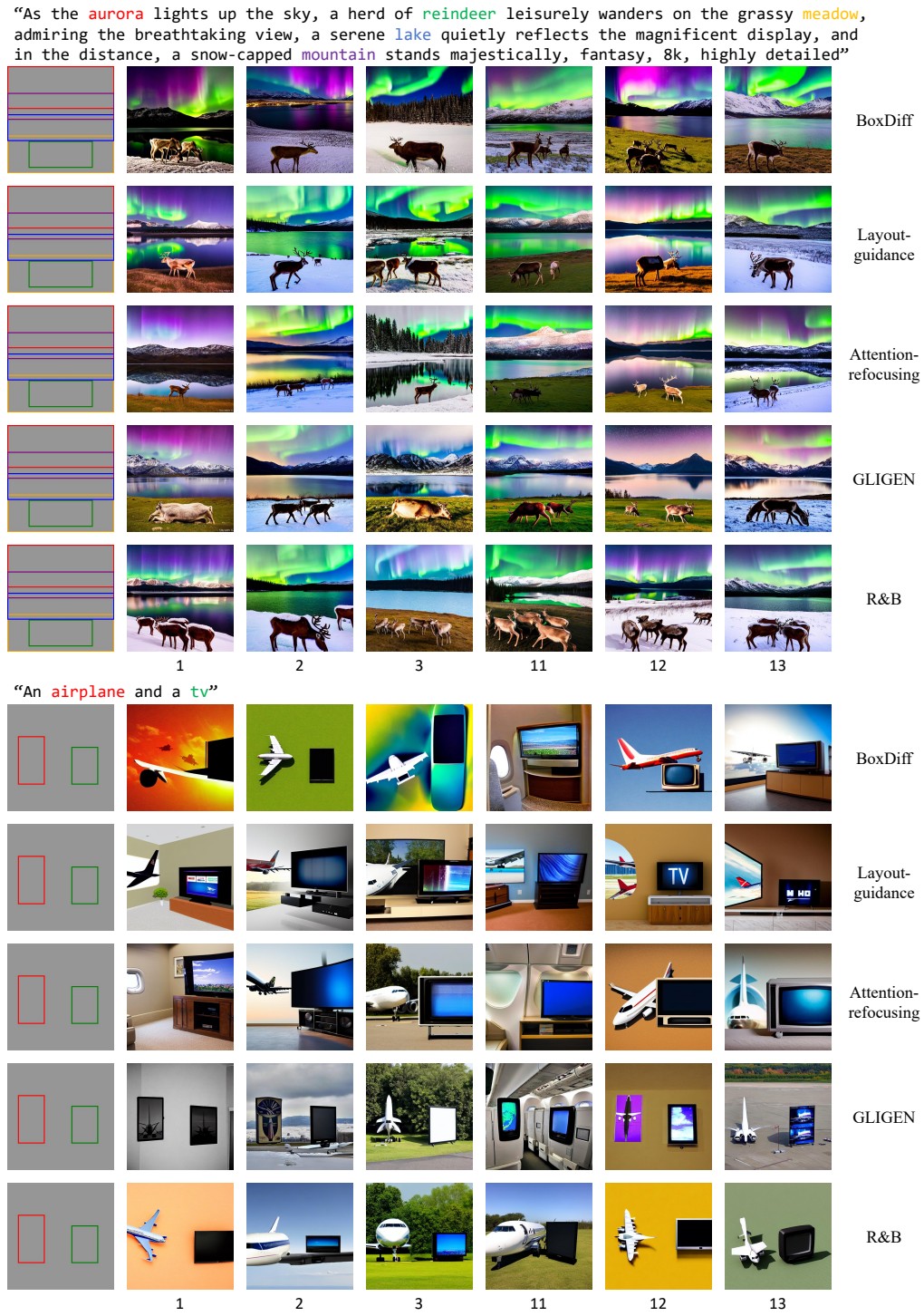

Figure 16: Visual results between training-free methods and training-based method. The example prompts are borrowed from the paper of BoxDiff. Seeds for latent initialization are annotated at the bottom. Zoom in for better view.

combined with R&B, the performance of GLIGEN gets consistently boosted (the rightmost column), because the region-aware loss further enhances the alignment between generative objects and the corresponding boxes, and the boundary-aware loss helps with the expression of different objects in their corresponding regions to better cooperate with the textual semantics. The performance of a

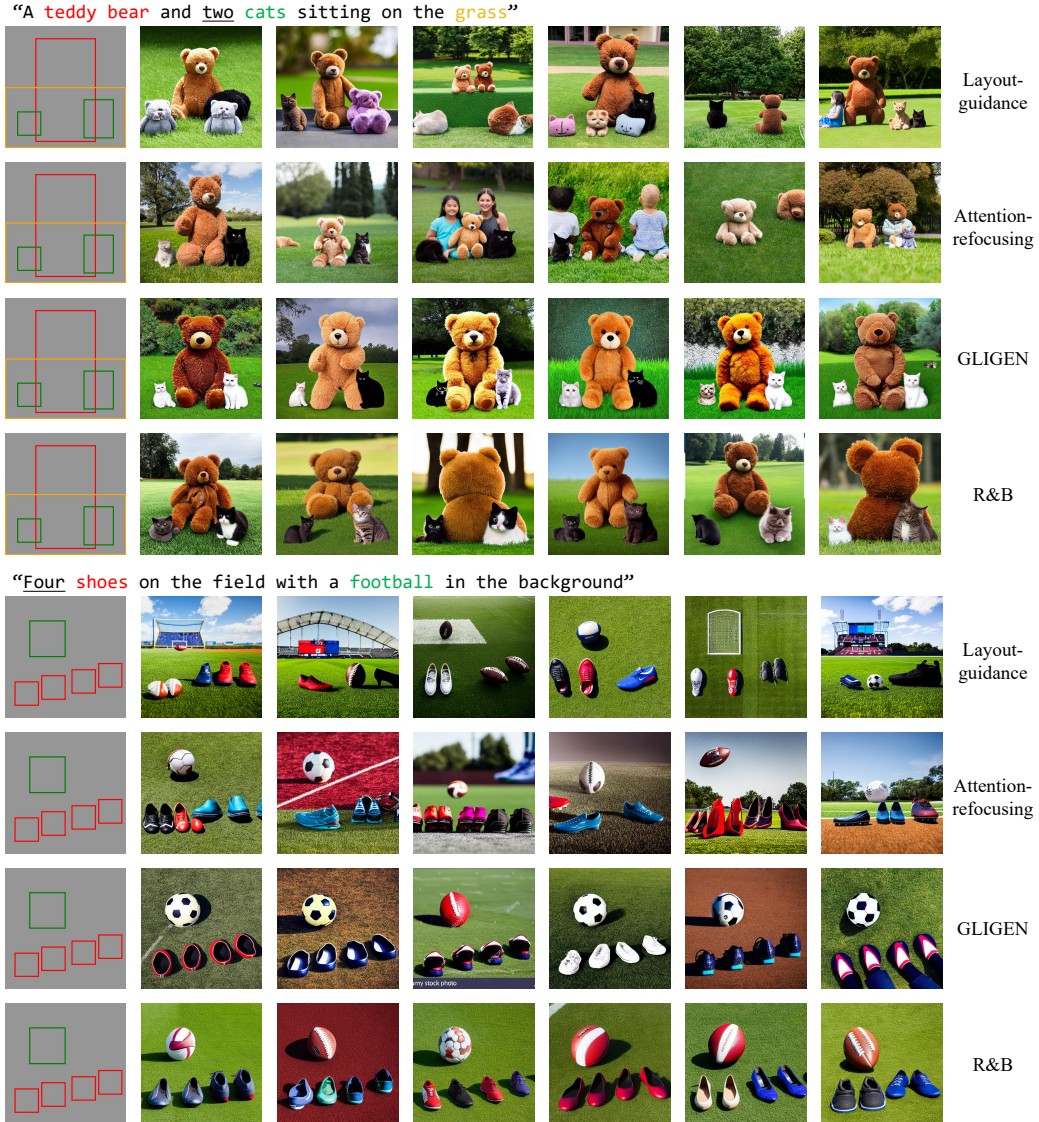

Figure 17: Visual results between training-free methods and training-based method on object counts setting. Zoom in for better view.

strong competing method "GLIGEN+SAR&CAR" is also reported in the second rightmost colmun of Table 6. SAR&CAR is the loss function adopted from Attention-refocusing (Phung et al., 2023).

**Qualitative.** Some visual comparisons between GLIGEN and our proposed R&B are shown in Figure 16, 17 and 18. The box information and textual prompts of Figure 16 are inherited from the paper of BoxDiff (Xie et al., 2023). The random seeds for latent initialization are shown in the bottom of the images. In the top half of Figure 16, all methods present relatively delicate outcomes, because the textual input is rather concrete and constitutes an in-domain prompt. In the bottom half of Figure 16, given an unusual prompt, we witness entanglement of semantics in the images generated by previous methods. For example, BoxDiff, Attention-refocusing and GLIGEN generate televisions inside the cabin of an airplane, Layout-guidance and GLIGEN generate TVs with airplane on the screen. By comparison, with the aid of the design region and boundary aware constraints, our approach better cope with the semantic confusion issue in grounded generation, and generate images that align with the layout instructions.

Comparisons on the setting of object counts are presented in Figure 17. BoxDiff is not included since it does not support the input format of multiple boxes pointing to the same object. Training-free

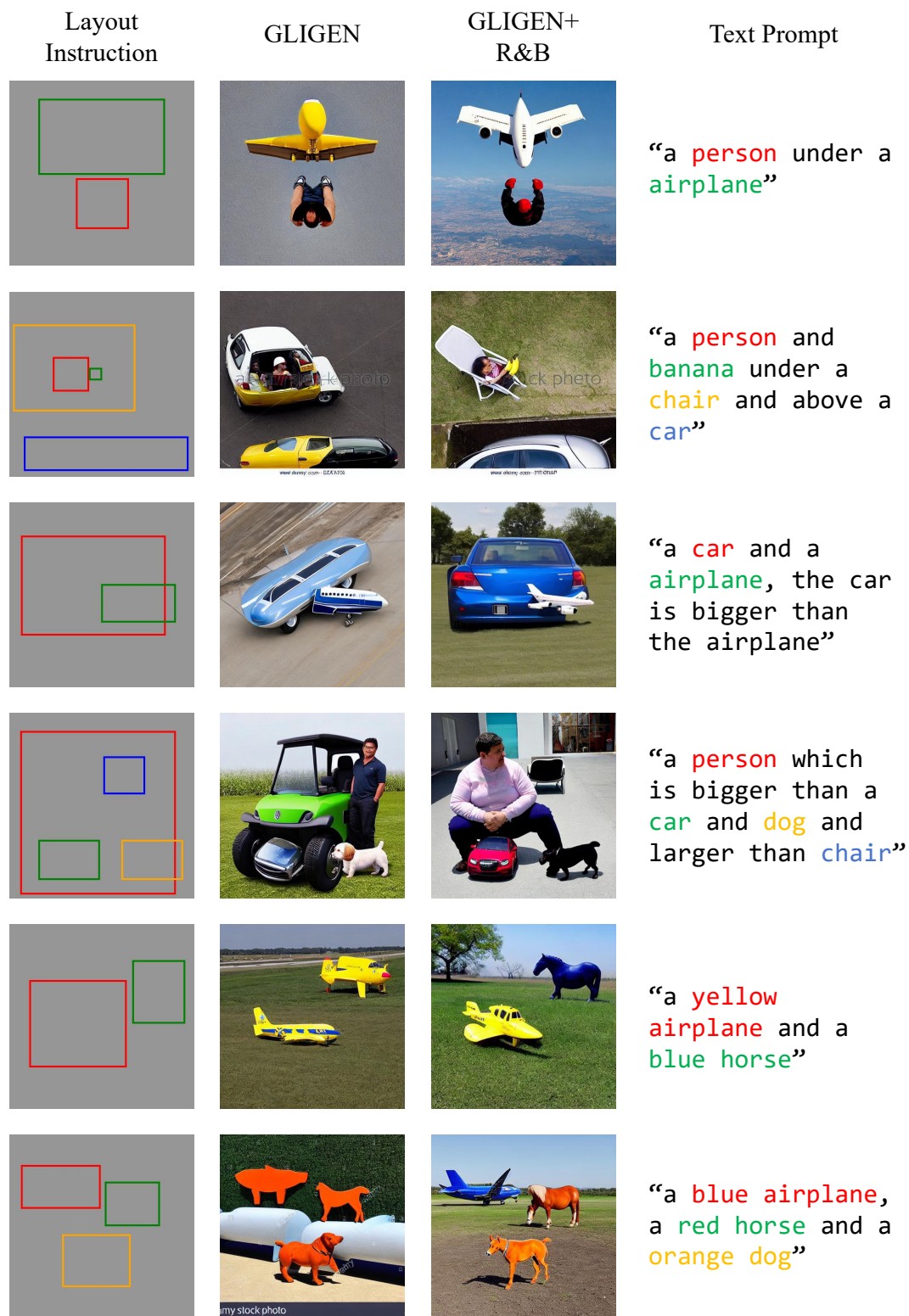

Figure 18: Visual results of the boost of GLIGEN combined with R&B on HRS benchmark. The uppermost, middle, lowermost two rows correspond to the spatial, size and color category, respectively.

methods like Layout-guidance and Attention-refocusing suffer from attribute leakage and missing object problems, therefore fail to consistently generate accurate object counts. Specifically, Layout-guidance generates cats that severely inherit features from the teddy bear and Attention-refocusing often forgets to generate small cats. The same problem also happens at the case of football and shoes. GLIGEN, as a training-based method, generates right amount of objects, but somewhat at the cost of decreased generative quality and diversity: (1) the lighting and texture of teddy bears and cats do not match the background, and (2) the teddy bear always faces forward and the four shoes share the same appearance. In comparison, our R&B generates consistently high-quality results while preserving diversity, due to the accurate layout guidance provided by the proposed two losses. Please note that the ambiguity of the word football causes two different appearances of the generated ball.

Figure 18 includes some visual examples on the HRS benchmark. We witness that GLIGEN provides strong prior to grounded multi-object generation, but still fails to accurately generate images with some unusual prompts. As an attention manipulation approach, R&B (1) boosts the generative fidelity

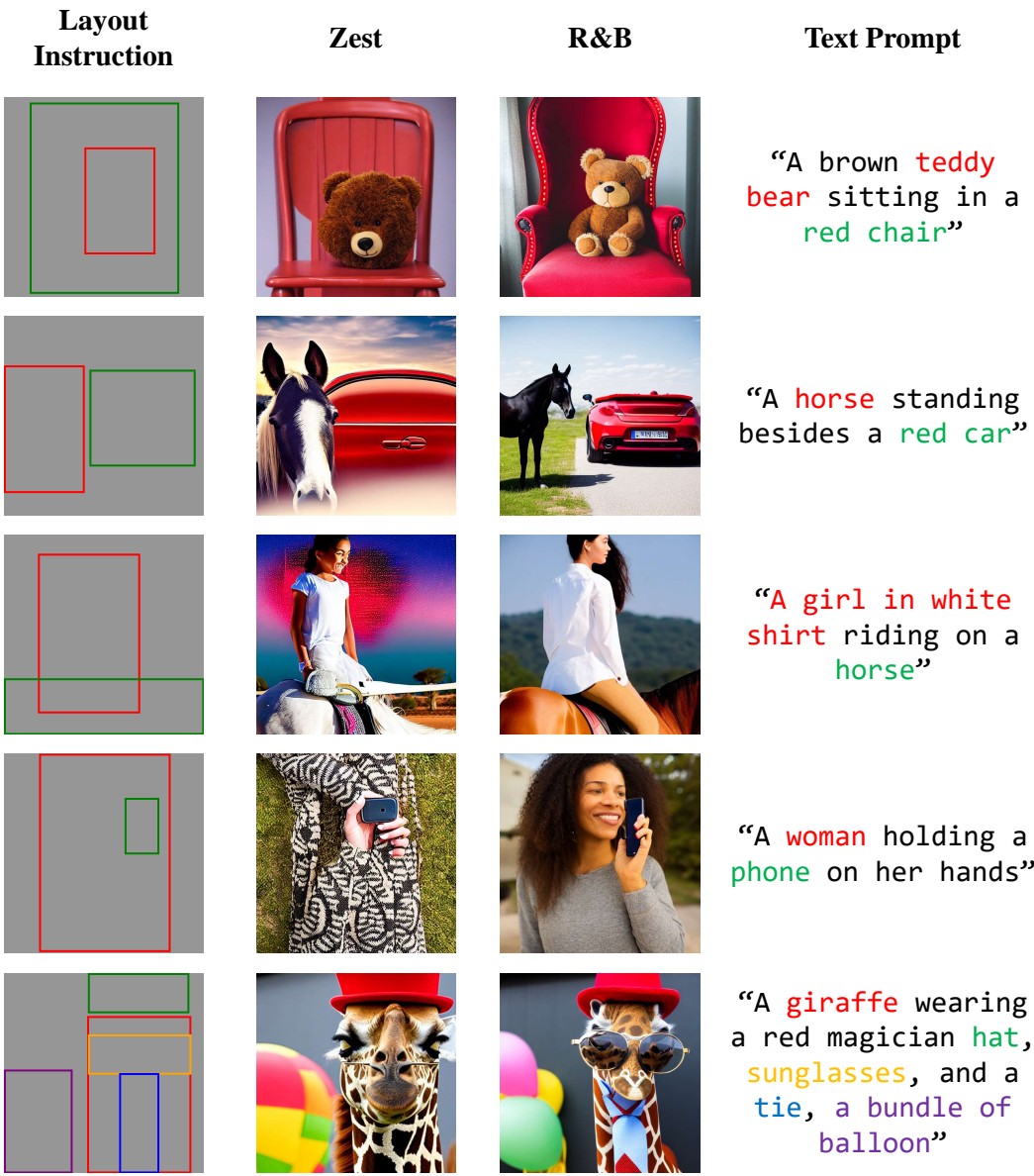

Figure 19: Visual comparisons between Zest and our proposed R&B.

of GLIGEN (1st row) of images, (2) helps to cope with the entanglement of different concepts (2nd row), (3) helps to fix poor semantics like size (4th row) and color (5th and 6th rows). This is confirmed by the quantitative results in Table 6.

## J    COMPARISON WITH ZEST

Zest (Couairon et al., 2023) is a zero-shot segmentation guidance approach for T2I diffusion models. It adopts a binary cross-entropy (BCE) loss for each cross-attention map entry, enabling the representation of objects in their respective mask regions, formulated as below:

$$\mathcal{L}_{\text{Zest}} = \sum_{i=1}^{K} (\mathcal{L}_{\text{BCE}}(\hat{\mathbf{S}}_i, \mathbf{S}_i) + \mathcal{L}_{\text{BCE}}(\frac{\hat{\mathbf{S}}_i}{\|\mathbf{S}_i\|_{\infty}}, \mathbf{S}_i)) \tag{19}$$

In Figure 19 we compare our method with Zest from the visual aspect. We observe when conditioned on grounded input like bounding boxes, which only coarsely describe the size and location of corresponding objects, Zest tends to impose too strong constraints to the diffusion models. As shown in the 2nd and 4th rows, the generated horse, car, and woman simply fill the entire boxes, presenting poor fidelity. We also observe the presence of peculiar artifacts on some synthetic images (2nd and 3rd rows). By comparison, our proposed R&B aligns well with the grounding input, and meanwhile presents images with better quality and semantic consistency.

