# OpenReview forum: "R&B: Region and Boundary Aware Zero-shot Grounded Text-to-image Generation"
_ICLR.cc/2024/Conference — ICLR 2024 poster_

### Official Review · Reviewer_F2Yp · 2023-10-30

**Soundness:** 3 good
**Presentation:** 3 good
**Contribution:** 3 good
**Rating:** 6
**Confidence:** 3

**Summary:**

This paper proposes R&B, a region- and boundary-aware cross-attention guidance for a zero-shot grounded Text-to-Image diffusion model. The goal of such a method is to produce an image conditioned on an input text prompt and bounding boxes for objects to be generated. While previous methods have tackled this task, they still lack in terms of accurate spatial generation and such issues such as missing objects and attribute-noun binding issues. First, attention maps for every concept are extracted. Next, bounding boxes are estimated from those attention maps and used for optimization in consecutive timesteps. The region-aware loss maximizes IoU of estimated boxes with ground-truth boxes, while the boundary-aware loss maximizes activation per object within the corresponding box.

**Strengths:**

- The paper is well-written and well-structured.
- The presented results are good.
- The proposed method is novel.

**Weaknesses:**

- It is unclear to me how the presented work should be viewed in comparison with ZestGuide [1]. The authors cite it and use the method in [1] to compute an aggregated attention map. However, there is no visual/quant. comparison and a discussion on the strengths/weaknesses are missing. From a high-level, it seems that [1] is able to solve the same issues using a simpler method, namely a zero-shot segmentation approach to guide the diffusion process on a per-object level. Furthermore, [1] is able to use free-form masks instead while the presented approach is limited to boxes.

[1] Couairon et al., 2023, Zero-shot spatial layout conditioning for text-to-image diffusion models

**Questions:**

- There is recent work that tackles poor attribute-noun binding in Text-to-Image models. Can the authors provide examples where one object has an additional attribute such as a color and visualize how the generated image is changed when varying the attribute or noun (e.g. "blue car" -> "orange car" -> "orange fox")? See [2] examples.

[2] Compositional Text-to-Image Synthesis with Attention Map Control of Diffusion Models, https://arxiv.org/abs/2305.13921

---

> ### Author Response · Authors · 2023-11-19
> **Response to Reviewer F2YP**
>
> _We thank the reviewer very much for the positive review and valuable suggestions. We are pleased that the reviewer thinks our paper is well written, the presented results are good and our method is novel. Below are our response to your questions, please kindly check them. **The changes in our revision are highlighted in shallow purple.**_
>
> **Q1: It is unclear to me how the presented work should be viewed in comparison with ZestGuide [1]. The authors cite it and use the method in [1] to compute an aggregated attention map. However, there is no visual/quant. comparison and a discussion on the strengths/weaknesses are missing. From a high-level, it seems that [1] is able to solve the same issues using a simpler method, namely a zero-shot segmentation approach to guide the diffusion process on a per-object level. Furthermore, [1] is able to use free-form masks instead while the presented approach is limited to boxes.**
>
> A1: Firstly, bounding boxes and segmentation maps are two forms of spatially conditioning input at different granularity. Bounding boxes are coarse but more user-friendly, while segmentation maps provide more spatial information but are harder to acquire. We think both two settings have research and application value. Secondly, we compare **ZestGuide** with our proposed **R&B** qualitatively in **Appendix L**. We observe when conditioned on grounded input like bounding boxes, which only coarsely describe the size and location of corresponding objects, Zest tends to fill the entire boxes, leading to poor generative results. Please refer to the paper for more analysis.
>
> **Q2:
> There is recent work that tackles poor attribute-noun binding in Text-to-Image models. Can the authors provide examples where one object has an additional attribute such as a color and visualize how the generated image is changed when varying the attribute or noun (e.g. "blue car" -> "orange car" -> "orange fox")? See [2] examples.**
>
> A2:
> We conduct the visual experiments following your suggestion in **Appendix I (Figure 16)**, illustrating how the proposed **R&B** copes with the attribute binding issues. Given consistent layout, we vary the corresponding phrases to adopt different visual outcomes. Results show that **R&B** helps to bind the nouns and their corresponding attributes across different variations with good robustness.

---

> > ### Comment · Reviewer_F2Yp · 2023-11-22
> > **Response to Authors**
> >
> > I thank the authors for their answers, explanations and additions to the paper. After reading the author's response and other reviews, I have decided to keep my score and tend towards acceptance.
> >
> > I would really appreciate details and if possible code on the reproduction of ZestGuide, given that the official implementation (to my knowledge) is not available.

---

> ### Author Response · Authors · 2023-11-22
> **Response to Reviewer F2Yp**
>
> Thank you for reading our revised manuscript and responses. We update the supplementary material and add the code to reproduce ZestGuide. For the reason that we do not find the official implementation of ZestGuide, so we reproduce its results under zero-shot grounded generation setting based on our understanding of the paper. We will also modify the relevant code and incorporate it into our final released code base as a competing baseline.
>
> Thank you again for valuable suggestions on improving the overall quality of our paper.

---

### Official Review · Reviewer_cFE6 · 2023-10-31

**Soundness:** 3 good
**Presentation:** 3 good
**Contribution:** 3 good
**Rating:** 6
**Confidence:** 4

**Summary:**

This paper introduces an approach to zero-shot grounded Text-to-Image (T2I) generation in diffusion models, termed Region and Boundary (R&B) aware cross-attention guidance for layout generation. Notably, this approach is training-free. By incorporating region-aware loss for guiding cross-attention maps and boundary-aware loss for refining localization, the method delivers enhanced ground-alignment performance. Empirical evidence from experiments and ablation studies underscores the method's superiority over existing state-of-the-art techniques for grounded T2I generation in diffusion models, demonstrating both qualitative and quantitative improvements.

**Strengths:**

1. This paper employs a dual approach, utilizing both region-aware loss and boundary loss to align the generated images with the text and boundary, presenting a rational and innovative methodology.

2. By incorporating dynamic thresholding into the aggregated cross-attention maps, this method introduces a novel optimization technique for accentuating foreground objects, enhancing the utility of aggregated cross-attention maps.

3. The visual results presented in this paper provide compelling evidence that the R&B method surpasses current state-of-the-art techniques in grounded Text-to-Image generation for diffusion models.

**Weaknesses:**

1. While the experimental results presented appear less solid, it's worth noting that Attention-refocusing is a versatile component that has demonstrated effectiveness in enhancing various methods, including Stable Diffusion, Layout-guidance, and GLIGEN. The results, as detailed in Tables 2 and 3 of the Attention-refocusing method, highlight the superiority of GLIGEN+CAR&SAR in achieving state-of-the-art (SOTA) performance, which outperforms the proposed method, as indicated in Table 1. Further clarification regarding this observation would be greatly appreciated.

2. Although the LIMITATIONS AND DISCUSSION section acknowledges that the model's capacity may be exceeded when dealing with a high number of objects, providing a more in-depth analysis of this phenomenon would enhance the discussion. Notably, the BoxDiff model appears to yield more favorable results in generating a large number of objects.

3. To enhance the comprehensiveness of the study, including additional visual comparisons with other methods would be advantageous. For instance, an evaluation of how our model performs with consistent boundary and prompts when compared to BoxDiff or other comparable methods could provide valuable insights.

4. It is imperative to incorporate more detailed analyses, such as color and object counts. Additionally, there is a concern regarding the last row in the rightmost image of Figure 1, where the text "A bundle of" does not seem to be adequately performed.

**Questions:**

1. How does this model perform when provided with more intricate prompts, such as "a pink rabbit with a white head," with the boundary pointing to both the "rabbit" and "head"?

2. I find the result in the last row of the rightmost image in Figure 4 perplexing. The image seems to exhibit incomplete deformation. What factors may have led to this particular outcome?

---

> ### Author Response · Authors · 2023-11-19
> **Response to Reviewer cFE6 #1**
>
> _We thank the reviewer very much for the positive review and valuable suggestions. We are pleased that the reviewer understands the innovation of our paper and thinks that the presented results are good and compelling. Below are our response to your questions, please kindly check them. **The changes in our revision are highlighted in shallow purple.**_
>
>
> **Q1: While the experimental results presented appear less solid, it's worth noting that Attention-refocusing is a versatile component that has demonstrated effectiveness in enhancing various methods, including Stable Diffusion, Layout-guidance, and GLIGEN. The results, as detailed in Tables 2 and 3 of the Attention-refocusing method, highlight the superiority of GLIGEN+CAR&SAR in achieving state-of-the-art (SOTA) performance, which outperforms the proposed method, as indicated in Table 1. Further clarification regarding this observation would be greatly appreciated.**
>
> A1:
> Thanks for the valuable suggestion. Quantitatively, our method still consistently surpasses the baseline methods (even equiped with **SAR&CAR**) on the benchmarks under training-free setting. We also compare with **SAR&CAR** under training-based setting. Details can be found in **Table 7** of **Appendix J**. We found that the strong baseline **GLIGEN+SAR&CAR** beat our method on spatial and size categories, but solely using **R&B** beats **GLIGEN+SAR&CAR** on the color category, which indicates that our method greatly enhances the semantic consistency of Stable Diffusion. Further, when equiped with **R&B**, GLIGEN gets greatly boosted and yeilds best performance on all the categories.
>
> **Q2: Although the LIMITATIONS AND DISCUSSION section acknowledges that the model's capacity may be exceeded when dealing with a high number of objects, providing a more in-depth analysis of this phenomenon would enhance the discussion. Notably, the BoxDiff model appears to yield more favorable results in generating a large number of objects.**
>
> A2: Generating multiple objects is challenging for T2I models due to the problem of object dominance, where the saliencies of objects compete with each other and some objects are hindered. In grounded generation setting, object saliencies are encouraged within the specified bounding boxes, mitigating the object dominance problem to some extent. Intuitively, the more accurate the spatial guidance is, the better performance on multiple objects will be.  Notably, this problem is alleviated when multiple objects and their box instructions have strong semantic consistency, i.e. the composition of these objects are common, which explains the delicate results of multiple-object examples presented in **BoxDiff**. Some of the visual examples from our revised manuscript include multi-objects, for example, **Figure 1 (last column)**, **Figure 11-13**, **Figure 13**, **Figure 18 (upper half)**, **Figure 20**, **Figure 22 (bottom half)**, **Figure 23 (bottom half)**, and **Figure 24 (last row)**. We run the same examples of BoxDiff in **Figure 17** for comparison. We use same random seeds (annotated at the bottom) for different methods. Our proposed **R&B** shows comparable or even better generative results. HRS dataset consistutes of compositions of 2, 3 and 4 objects. The proposed R&B method surpasses all training-free methods including BoxDiff on the HRS dataset, proving the ability of handling multiple objects of R&B. R&B provides more accurate spatial guidance through the proposed two losses, leading to the performance gain.
>
>
>
> **Q3:
> To enhance the comprehensiveness of the study, including additional visual comparisons with other methods would be advantageous. For instance, an evaluation of how our model performs with consistent boundary and prompts when compared to BoxDiff or other comparable methods could provide valuable insights.**
>
> A3:
> We conduct a qualitative comparison using cases extracted from the **BoxDiff** paper between our proposed **R&B** and other methods in **Figure 17**. Please refer to **Appendix J** 'Qualitative' part for more analysis. We fix the random seeds and annotated them below the images for better comprehension. We also observe that the generative layouts of **BoxDiff** are unstable and sensitive to the initial latents due to the top-k optimization strategy. On the contrary, our method show robustness to different prompts and seeds.

---

> ### Author Response · Authors · 2023-11-19
> **Response to Reviewer cFE6 #2**
>
> **Q4: It is imperative to incorporate more detailed analyses, such as color and object counts. Additionally, there is a concern regarding the last row in the rightmost image of Figure 1, where the text "A bundle of" does not seem to be adequately performed.**
>
> A4: For object color, we present additional qualitative comparisons in **Figure 22** and **Figure 23**, showing that our propsed **R&B** is able to handle attribute binding correctly and generate the color according to text prompt, even when the attribute is not common. We also show the robustness of our method by varying nouns and attributes (e.g., color) in **Appendix I (Figure 16)**. For object counts, we show visual comparison in **Figure 18**, we witness that the spatial accuracy of images generated by our method is even comparable with the training-based method. Please refer to **Appendix J** 'Qualitative' part for more analysis. Our proposed layout constraints not only provide accurate layout guidance, but also cope with semantic issues simultaneously, which helps the model to bind attributes with right nouns effectively, and helps to generate objects with correct number according to the layout instructions. As for the question that the text "A bundle of" does not seem to be adequately performed, we find that when the bounding box is associated with only the word "balloon," the model tends to generate only a single balloon inside the box. One simple solution is to incorporate 'a bundle of' into the object phrase, letting spatial guidance imposing on the whole phrase, and multiple balloons are observed. You can refer to **Figure 1 (last row, last column)** and **Figure 24 (last row, last column)** , by changing the input form, we can intuitively observe that the semantics of "A bundle of" can be adequately performed.
>
> **Q5: How does this model perform when provided with more intricate prompts, such as "a pink rabbit with a white head," with the boundary pointing to both the "rabbit" and "head"?**
>
> A5: We present visual results on intricate prompts in **Appendix J**, where the prompt "a pink rabbit with a white head" is included. Three more similar intricate prompts are provided for visual comparison. Please refer to **Appendix K** 'Performance on intricate prompts' part for detailed discussion.
>
> **Q6: I find the result in the last row of the rightmost image in Figure 4 perplexing. The image seems to exhibit incomplete deformation. What factors may have led to this particular outcome?**
>
> A6: We think the model tries to synthesize view from the airplane porthole since the prompt specifies background as runway. Yet finally only a portion of the windows is generated, resulting in strange distortions in the upper half. The regions of the "running horse" guided by layout constraints do not exhibit any undesirable distortions. Commonly, when the layout guidance completely fails, the appearance of images may seem like **Figure 4 (last row, column 3)**, which seems notably out-of-distribution.

---

### Official Review · Reviewer_KFuf · 2023-10-31

**Soundness:** 3 good
**Presentation:** 2 fair
**Contribution:** 2 fair
**Rating:** 6
**Confidence:** 4

**Summary:**

The paper proposes a new method for grounded text-to-image generation based on the introduced region-aware and boundary-aware losses. These two losses help to modify the generation process of a pre-trained diffusion model in a way that generated objects adhere to the locations of provided bounding boxes. The method is compared to recent state-of-the-art backward methods such as BoxDiff, Attention refocusing, Layout-guidance. The experiments demonstrate that the method outperforms previous approaches in the accuracy of following the conditioning instructions.

**Strengths:**

- The paper addresses an interesting and in-demand problem of grounded T2I generation with diffusion models, tailored for applications that require control over locations of objects via user-friendly inputs.
- The proposed approach is compared to the most recent backward baselines, including BoxDiff [ICCV23].
- The demonstrated results indicate clear improvement over the baselines in terms of accuracy of following the conditioning instructions like the size or location of specified objects.

**Weaknesses:**

- My main concern is unclarity in the conceptual differences of the proposed method to other methods. Training-free modification of attention map is not a new idea as other methods like (Chen et al., 2023; Xie et al., 2023; Phung et al., 2023) also design objectives that encourage the model to shift objects towards specified bounding boxes. These methods generally exhibit same motivation (align objects with bounding boxes) with similar implementation (shift cross-attention maps). It is not clear what makes the proposed approach conceptually different and where the performance gain comes from.
   - Could the authors please elaborate their explanations in intro: 1) "previous methods fail to provide accurate spatial guidance" - why does this happen based on their design, and why does this not happen in R&B method based on its design? 2) "inherit inconsistencies from the original T2I model" - doesn't this also happen in R&B? Or what part of the design helps to avoid them?
- Similarly, it is not clear to me what is the technical novelty of the proposed method. The narration of the method is structured as the adaptation of existing methods (Li et al., 2023b, Li et al., 2023b) to the task of interest without much new analysis. Could the authors please elaborate what could be a broader-impact technical lesson or insight for the community from Sec. 3?
- I feel the presentation of experiments in Sec. 4 can be improved. For example, there is no presentation of baselines used in Table 1, Fig. 4, so it is difficult to the match the names of methods with references to respective papers.
- Why are there no comparisons to some other baselines, like GLIGEN?

**Questions:**

Please answer the questions or comment on concerns from the Weaknesses section.

[UPD rebuttal - score raised from 5 to 6]

---

> ### Author Response · Authors · 2023-11-19
> **Response to Reviewer KFuf #1**
>
> _We thank the reviewer very much for the valuable suggestions. We are pleased that the reviewer thinks we are addressing an interesting and in-demand problem, and we compare with recent works comprehensively, indicating clear improvement over existing methods. Below are our responses to your comments. **The changes in our revision are highlighted in shallow purple.**_
>
> **Q1: Unclarity in the conceptual differences of the proposed method to other methods.**
>
> A1: Most diffusion-based T2I models inject external textual condition into the generative process through the cross-attention, the most effective approach for controllable generation with diffusion models is to leverage cross-attention manipulation. From this perspective, most works share the same idea to achieve zero-shot grounded generation, that is, to align the cross-attention maps of diffusion models with the grounding input. In this work, our motivation is primarily based on an observation of the two critical issues previous zero-shot grounded generation methods face with: (1) previous methods fail to provide accurate spatial guidance, (2) they inherit inconsistencies from the original T2I model. Detailed discussions on previous methods **Layout-guidance** [1], **BoxDiff** [2], and **Attention-refocusing** [3] can be found in the **Appendix B**.
>
>  **As for the spatial accuracy**, **BoxDiff** and **Attention-refocusing** borrow insights from previous work [4] to amplify the top values of cross-attention maps within the bounding boxes, and reduce the top values of the outer regions to control the generative layout of different concepts. Yet the min-max optimization makes the layout guidance unstable and sensitive to the starting latents. See **Figure 9**, **Figure 17 (bottom half)**, **Figure 18** for example, we observe that some samples align with the box constraints, while others present inaccuracy of shape and localization. And optimizing top values only ensures that the most discriminative parts of the object are contained within the bounding boxes, while other parts of the object may not adhere to the layout constraints. The score function of **Layout-guidance** encourages the concentration of attention within the bounding boxes, which may lead to tricky solution, i.e., the objects are much smaller than the boxes, but the value of the score function are close to 0. Examples can be found in **Figure 4 (column 5, row 4)**, where the cup is much smaller than the box, **Figure 9 (column 3 and 6, row 4)**, where the globe is much smaller than the box. Moreover, when peak attention values appear within the bounding boxes, the value of score function experiences a sudden drop, which may affect the accuracy of guidance. See the upper half of **Figure 9 (row 3)**, the trunk is within the bounding box but the branches and leaves are outside the box. Quantitative results (**Table 5 in Appendix F**) on box alignment are consistent with aforementioned discussions.
>
> **As for the semantic consistency**, one key insight is that if the attention map is too smooth, the object may lack discriminative regions, which can lead to missing objects or attribute leakage. Thus, previous works [4,5] essentially increasing the variance of cross-attention maps to tackle the semantic issues. Both **Layout-guidance** and **BoxDiff** enhance the response of cross-attention within the layout equally, which do not effectively boost the total variance, thus lead to missing objects (**Figure 4, 9** and **18**) and concept fusion (**Figure 9** and **18**). The self-attention refocusing introduced by **Attention-refocusing** aims to ensure semantic consistency within the bounding box but does not consider the conditional textual information, it may have negative effects on guidance when the generative layout deviates significantly from the ground truth, resulting in tough semantic issues (**Figure 4, 9** and **18**). Results in **Table 1** in **Section 4.2** reveal that previous methods suffer from fatal semantic issues.

---

> ### Author Response · Authors · 2023-11-19
> **Response to Reviewer KFuf #2**
>
> (**Continue from A1**) By comparison, by directly addressing two key issues (spatial, semantic), our proposed **R&B** achieves significant performance improvements. (1) The region-aware loss aims to bridge the gap between the consecutive attention maps and the binary masks. Different from previous methods, we directly model the divergence between generative layout and ground truth by dynamic thresholding and box selection, and gradually optimize the predicted boxes towards the target boxes. The IoU between pseudo boxes and ground truth modulates the guidance scale, which mitigates the tricky solution encountered in **Layout-guidance**. The discrete sampling makes each foreground elements contribute equally to the score function, avoiding the affect of peak response.  Hence, the generative outcomes better align with the ground truth layout. (2) The boundary-aware loss helps refine boundaries of attention maps in object regions, this essentially enhance the variance within the layout, and enhances the semantic consistency to the textual prompts, and further ensures adherence to layout constraints. We achieve high spatial alignment and semantic consistency simultaneously, making our method distinct from previous methods.
>
> **Q2: The technical novelty of the proposed method.**
>
> A2: Some technical novelties can be found in the summary of our method in A1. The previous work **Divide&Bind** [5] brings a valuable insight that enhancing the variance of cross-attention can effectively address semantic issues related to the corresponding concepts. Inspired by this, we design the boundary-aware loss for zero-shot grounded generation. Indeed, our common goal is to address semantic issues, but from the implementation perspective, we are fundamentally different from [5]. [5] simply enhances the overall variance of the cross-attention map. In contrast, our method adopts and sharpens the Sobel boundaries within the boxes while reducing the variance of objects outside the boxes. Additionally, our guidance is modulated by the predicted IoU to prevent excessive optimization that could impact accuracy and quality. Furthermore, our method employs two loss functions, which can not be structured as an adaption of [5].
>
> In summarry, (1) the clear motivation of the proposed two layout constraints (region-aware loss for layout accuracy, boundary-aware loss for semantic consistency), (2) the exhaustive analysis and discussions on the effects of each operation in **Section 3**, illustrated in **Section 4.3**, **Appendix C** and **E**  could have a impact for the community from the technical perspective. Further, and the promising improvement over existing methods both quantitatively and qualitatively (even comparable with training-based methods) and the vital problem (grounded generation without any data) may have a broader-impact for controllabe image generation with user-interface.
>
> **Q3: The presentation of experiments in Sec. 4 can be improved.**
>
> A3: This is a good advice, we present the details of different competing methods in **Appendix B** of the original manuscript, but we do not indicate readers in the main paper to refer to the Appendix. In the revised manuscript, we add a brief hints in **Section 4.1** to guide readers towards the content in the appendix, with detailed discussions of different methods.
>
> **Q4: Why are there no comparisons to some other baselines, like GLIGEN?**
>
> A4: We consider our method to be applicable without training data, and therefore, we compare it with other methods under the same setting. However, we notice that many reviewers are interested in a comparison with training-based methods. In response, we include comparison with a typical training-based method, **GLIGEN** [6], in the revision to enhance the completeness of the paper. We add quantitative and qualitative comparisons with **GLIGEN** in **Appendix J**. You can see **Figure 17-19**, **22**, **23** and **Table 7** in our revised version for details of comparison results and discussions.
>
>
> [1] Chen, Minghao, Iro Laina, and Andrea Vedaldi. "Training-free layout control with cross-attention guidance." arXiv preprint arXiv:2304.03373 (2023).
>
> [2] Xie, Jinheng, et al. "Boxdiff: Text-to-image synthesis with training-free box-constrained diffusion." Proceedings of the IEEE/CVF International Conference on Computer Vision. 2023.
>
> [3] Phung, Quynh, Songwei Ge, and Jia-Bin Huang. "Grounded Text-to-Image Synthesis with Attention Refocusing." arXiv preprint arXiv:2306.05427 (2023).
>
> [4] Chefer, Hila, et al. "Attend-and-excite: Attention-based semantic guidance for text-to-image diffusion models." ACM Transactions on Graphics (TOG) 42.4 (2023): 1-10.
>
> [5] Li, Yumeng, et al. "Divide & bind your attention for improved generative semantic nursing." arXiv preprint arXiv:2307.10864 (2023).
>
> [6] Li, Yuheng, et al. "Gligen: Open-set grounded text-to-image generation." Proceedings of the IEEE/CVF Conference on Computer Vision and Pattern Recognition. 2023.

---

> ### Comment · Reviewer_KFuf · 2023-11-21
> **Thanks for getting back**
>
> I thank the authors for their answers and explanations.
>
> After reading the response and other reviews, I find that the visual performance of the method and comparisons to recent baselines are the most strong parts of the submission. I would evaluate the clarity of explanations and presentation as a downside, but this was majorly addressed in the rebuttal.
>
> I would suggest the paper to be accepted, but I encourage the authors to incorporate the rebuttal and explanation to the main paper, and maybe do one more re-writing pass to ensure that the global picture and conceptual differences are clear for readers.

---

> > ### Author Response · Authors · 2023-11-21
> > **Thank you for reviewing our response and increasing the score**
> >
> > We sincerely appreciate your thorough review of our revisions and responses, reevaluation of our method, and the decision to recommend acceptance of our paper. We will seriously take the suggestions and questions during the discussion process and carefully incorporate them into the main paper to ensure clarity regarding our insights and motivation. Once again, we would like to express our gratitude for your valuable time and contribution to the discussion, which has significantly helped improve the overall quality of our paper.

---

### Official Review · Reviewer_Nptk · 2023-11-01

**Soundness:** 3 good
**Presentation:** 3 good
**Contribution:** 3 good
**Rating:** 6
**Confidence:** 4

**Summary:**

This paper presents a training-free approach to grounded text-to-image generation. The key idea is to instrument the cross-attention weights in a pre-trained Stable Diffusion model to control image layout, thereby generating objects within the given bounding boxes. The key innovation of the proposed method lies in the energy functions in guidance. Two such functions are introduced; the region-aware loss facilitates the minimum-sized box enclosing the object to match the ground-truth box, whereas the boundary-aware loss encourages rich content variation within a box. Both qualitative and quantitative experiments validate the effectiveness of the proposed loss functions. The method outperforms several training-free baselines by a wide margin, enabling more precise layout control over text-to-image generation.

**Strengths:**

- The method is training-free. It controls image layout by manipulating cross-attention weights using guidance. Compared to methods that train auxiliary modules for controllability, training-free methods readily support new architectures and model checkpoints without re-training.

- The proposed method carefully reasons about the failure cases of existing methods and introduces key improvements to cross-attention based guidance for grounded text-to-image generation. These improvements cover different aspects of object localization, including dynamic thresholding of attention maps, differentiable box alignment and and boundary awareness. They offer key insights into the internal workings of Stable Diffusion and may be of interest to the community in a broader context.

- The proposed loss terms effectively improve bounding box localization and address missing objects. The method outperforms several training-free baselines in both qualitative and quantitative experiments. Visualizations of cross-attention maps demonstrate clear localization and separation of object regions.

**Weaknesses:**

- Several training-free methods similarly apply cross-attention guidance to control object shape and location. To this end, the contribution of the present work is more or less incremental. While the loss functions are new, the novelty mainly lies in the implementation details, with the overall idea (cross-attention guidance) largely identical to previous methods.

- Comparison with training-based methods are lacking. For example, GLIGEN (which the paper cites) is a training-based method for grounded text-to-image generation. As they solve the same task, it would be helpful to compare with those methods qualitatively and quantitatively to reveal the strength and weakness of both approaches (and also for the sake of completeness).

**Questions:**

- I want the authors to comment on the runtime of the proposed method. How is it compared to the vanilla generation process (i.e., without guidance). In addition, how sensitive is the method to hyper-parameters?

- Is the method able to handle overlapping objects? Most examples in the paper show non-overlapping bounding boxes. I am curious about how cross-attention guidance behaves when two or more objects compete for the same pixel.

- I want to understand the performance of the method with respect to object size. Since the attentions maps have relatively low resolution (down to 16x16), I am wondering whether the losses (especially the boundary-aware loss) are still effective in the presence of small objects. Some qualitative results and visualizations would be helpful.

---

> ### Author Response · Authors · 2023-11-19
> **Response to Reviewer Nptk #1**
>
> _We thank the reviewer very much for the positive review and valuable suggestions. We are pleased that the reviewer acknowledges the practical significance of the task we have addressed. And the reviewer comments that our method effectively enhances the spatial accuracy and semantic consistency of T2I diffusion models, which may be of interest to the broader research community. Below are our responses to your comments. **The changes in our revision are highlighted in shallow purple.**_
>
> **Q1: While the loss functions are new, the novelty mainly lies in the implementation details, with the overall idea (cross-attention guidance) largely identical to previous methods.**
>
> A1: Indeed, cross-attention guidance is a routine (and the most effective way) to achieve controllable generation, for most diffusion-based T2I models inject external textual condition into the generative process through the cross-attention. In our work, we point out (**Section 1**) that existing training-free grounded generation methods suffer from two main critical issues: (1) they fail to provide accurate spatial guidance, (2) they inherit the semantic inconsistency issues from the original T2I model, and validate our observations via qualitative and quantitative experiments. Indeed, we are not the first to utilize cross-attention guidance to address zero-shot grounded generation. However, we tackle the two critical issues mentioned above by designing novel layout constraints accordingly (**Section 3**), and provide comprehensive discussions on the implementation (**Section 4.3** and **Appendix C**, **E**). As recognized by the reviewer, we address the practical problem of Stable Diffusion and greatly boost its performance on compositional generation. Compared to previous training-free methods, our approach exhibits a closer performance approximation to training-based methods that rely on extensive training. These may inspire future research in this domain, or even gain a broader impact on controllable generation with user interface. We also add discussions of different training-free method to our revised manuscript in **Appendix B**. Previous methods struggled to achieve spatial alignment and meanwhile addressing semantic issues, whereas our method successfully accomplishes both. This is further supported by exhaustive qualitative and quantitative experiments.
>
> **Q2: Comparison with training-based methods are lacking. e.g., GLIGEN.**
>
> A2: We thank the reviewer for the useful suggestion. For sake of completeness of our experiments, we add quantitative and qualitative comparisons with GLIGEN [1] in **Appendix J**. See **Figure 17-19**, **22**, **23** and **Table 7** in our revised version for details. One key observation is that training-based methods (like GLIGEN) tend to exhibit strong biases according to the training data, that is, it behaves well on common prompts (higher accuracy on spatial and size categories in **Table 7**), while fails to generate images properly on unusual prompts (poor accuracy on color categories in **Table 7**, semantically entangled concepts in the generated images). These implies that although well-trained on annotated data, GLIGEN still suffers from severe semantics issues inherited from the original T2I Diffusion model, and struggles to generate out-of-domain images (**bottom half of Figure 17**, **Figure 22-23**). Further, images generated by GLIGEN sometimes present undesirable watermarks inherited from the COCO training-set, for example, **Figure 18 (row 3 of the bottom half)**, **Figure 19 (row2 and row 6)**. These indicates that poisons in the training data may also harm the quality of training-based grounded generation methods.
>
> **Q3: Runtime of the proposed method.**
>
> A3: The number of guidance steps and optimization iterations per step are the critical factors that affect computational latency. By default, we adopt the DDIM scheduler with 50 denoising steps. We perform layout-guidance at the first 10 steps, and optimize the noisy latent for 5 iterations per step. Essentially, we add 50 forward processes and 50 backward process to the vanilla denoising procedure (50 forward steps). We list the practical runtime of several training-free methods mentioned in our manuscript on a single NVIDIA RTX-4090 GPU as below:
>
> | &emsp; Stable Diffusion &emsp; | Layout-guidance | &emsp; BoxDiff &emsp; | Attention-refocusing | &emsp; R&B &emsp;|
> | :------: | :---: |  :---: |  :---: |  :---: |
> |     7.9s    |    25.3s     |  31.6s|  51.4s | 29.7s  |
>
> Additional time cost is relatively acceptable because, with vanilla diffusion alone, it may not be possible to generate an image that aligns with the user's grounding input, even after generating dozens of images. Nonetheless, further analysis regarding the trade-off between guidance iterations and the image quality for zero-shot grounded generation is meaningful, and may be comprehensively discussed in our future work.

---

> ### Author Response · Authors · 2023-11-19
> **Response to Reviewer Nptk #2**
>
> **Q4: How sensitive is the method to hyper-parameters?**
>
> For training-free layout guidance methods, the selection of hyperparameters is quite complex. It is hard to find a set of hyperparameters that is suitable for all grounding inputs, textual prompts, and random seeds. Practically, we do not intentionally seek the optimal hyperparameters because within a reasonable range, many generated images are of good quality empirically. In the $\bf{9^{th}}$ page of our paper, we have numerically studied the effect of the scale of the cross-attention guidance scale. We further study the impact of hyper-parameters $\lambda$, $\lambda_s$ and $\lambda_a$ qualitatively and quantitatively. Details can be found in **Appendix H**. Visually, the outcomes within a particular parameter range (indicated by blue dashed boxes) are generally consistent with the bounding box instructions (**Figure 15**). We also discuss the impact of parameter $\lambda_s$ (with $\lambda=0.4, \lambda_a=1$) from a quantitative perspective in **Table 6**. Our default hyper-parameters configuration ($\lambda=0.4, \lambda_a=1, \lambda_s=1.5$) does not yield best alignment with the grounding input, but achieves a balance between spatial accuracy and semantic consistency.
>
> **Q5: Is the method able to handle overlapping objects?**
>
> Some of the visual examples in our original manuscript present objects with overlaps, **Figure 1 (columns 2-4)**, **Figure 3**, **Figure 6 (row 1)**, **Figure 8 (row 1-2)**, **Figure 11** and **Figure 13-14**. We also illustrate more samples in our revised version, see **Figure 17 (upper half)**, **Figure 18 (upper half)**, **Figure 24 (row 1, 4, and 5)**. Some hard cases are also presented, for example, **Figure 20** (overlaps between multi objects), **Figure 22-23** (intricate prompts). One intuitive attention map visualization is shown in **Figure 2** of **Section 3** with prompt "A handbag placed on the wooden bench", we observe that normalized attention map of "bench" is hollow because the corresponded region is overlapped by the "handbag". Detailed discussion can be found in **Appendix K**. When users provide reasonable grounding inputs, our method effectively activates the diffusion priors on object position and relations, resulting in desirable generative outcomes.
>
> **Q6: I want to understand the performance of the method with respect to object size.**
>
> Generating small objects is relatively difficult for the original Stable Diffusion, because the resolution of the noisy latents is 64$\times$64. In our implementation, we adopt the Stable Diffusion V-1.5 as our base model, which injects external conditional information with cross-attention at resolution 8, 16, 32, 64 respectively. Most cross-attention control methods [2, 3, 4] design score functions on the cross-attention maps of resolution $16\times16$, this may results in object shifts from the layout constraints when scaling up to the image space. As mentioned in **Section 3 (Page 4)**, we upscale the cross-attention maps to resolution $64\times64$ and aggregate them to a unified map, we then calculate the score functions for layout-guidance. This helps mitigate the aforementioned issue of object shift. Some visual examples on small ojbects are shown in our revised manuscript, and can be found in **Figure 18**, **Figure 22**, **Figure 23 (bottom half)**, **Figure 24 (row 4)**, we also accordingly discuss our performance on small objects in **Appendix K** and **Figure 21**. Generating grounded small ojbects is indeed a tough setting for training-free methods, although showing robustness in handling smaller boxes, our method fails when the size of given layout constraints is too small (**last column in Figure 21**).
>
> [1] Li, Yuheng, et al. "Gligen: Open-set grounded text-to-image generation." Proceedings of the IEEE/CVF Conference on Computer Vision and Pattern Recognition. 2023.
>
> [2] Chen, Minghao, Iro Laina, and Andrea Vedaldi. "Training-free layout control with cross-attention guidance." arXiv preprint arXiv:2304.03373 (2023).
>
> [3] Phung, Quynh, Songwei Ge, and Jia-Bin Huang. "Grounded Text-to-Image Synthesis with Attention Refocusing." arXiv preprint arXiv:2306.05427 (2023).
>
> [4] Xie, Jinheng, et al. "Boxdiff: Text-to-image synthesis with training-free box-constrained diffusion." Proceedings of the IEEE/CVF International Conference on Computer Vision. 2023.

---

> > ### Comment · Reviewer_Nptk · 2023-11-23
> >
> > Thanks for addressing my comments. Overall, this is a solid paper with strong empirical results. The revised draft looks good to me, and I would like to keep my rating.

---

> > > ### Author Response · Authors · 2023-11-23
> > > **Thank you for maintaining the score**
> > >
> > > Thank you very much for recognizing our work. We are really delighted that your valuable feedback helps improve the completeness of our work greatly, and has provided us with a lot of inspiration for our future work.

---

### Author Response · Authors · 2023-11-19
**General Comment**

We express our gratitude to all the reviewers for their comprehensive reviews and valuable questions. We are happy to hear that this paper addresses an interesting and in-demand problem (**Reviewer Nptk and KFuf**), the proposed layout constraints for zero-shot T2I generation is rational and novel (**Reviewer Nptk, cFE6, and F2Yp**), and performance improvements showed in experiments are promising with both qualitative and quantitative results (**All Reviewers**), we are also delighted to see that the reviewers have expressed a keen interest in exploring how R&B, as a user interface method, addresses challenging T2I tasks (**Reviewer Nptk and cFE6**).

We have carefully revised our paper according to the suggestions and questions from the reviewers (**marked in shallow purple**). The majority of the modifications have been included in the appendix, and can be summerized as below:

+ We have added a brief hint to **Section 4.1** so as to guide readers towards the content in the **Appendix B**, where we provide details and discussion of all the competing baselines (**Reviewer KFuf**).
+ We further discuss the impact of hyperparameters from the qualitative and quantitative perspectives in **Appendix H** (**Reviewer Nptk**).
+ We provide visual examples in order to demonstrate how our method can effectively tackle poor attribute-noun binding problems in T2I diffusion models in **Appendix I** (**Reviewer F2Yp**).
+ We comprehensively compare our approach with the training-based grounded generation method GLIGEN [1], both quantitatively and qualitatively, and provide corresponding analysis in **Appendix J** (**Reviewer Nptk, KFuf, and cFE6**).
+ We have expanded the results of our method under challenging prompts to provide a more comprehensive understanding in **Appendix K** (**Reviewer Nptk and cFE6**).
+ A related work Zest [2] on zero-shot segmentation-guided T2I generation is reproduced and discussed in **Appendix L** (**Reviewer F2Yp**).

We have responded in detail to each reviewer's questions below their respective reviews. Please kindly check them out.


[1] Li, Yuheng, et al. "Gligen: Open-set grounded text-to-image generation." Proceedings of the IEEE/CVF Conference on Computer Vision and Pattern Recognition. 2023.

[2] Couairon, Guillaume, et al. "Zero-shot spatial layout conditioning for text-to-image diffusion models." Proceedings of the IEEE/CVF International Conference on Computer Vision. 2023.

---

### Meta-Review · Area_Chair_EXEH · 2023-12-08

**Metareview:**

This paper proposes a novel approach to training-free spatial guidance for text-to-image diffusion models. The rebuttal addressed most review comments. Three of four reviewers responded to the author rebuttal and maintained or raised their score. All reviews recommend accepting the paper.

**Justification For Why Not Higher Score:**

All reviewers rate the paper as 6: "marginally above the acceptance threshold"

**Justification For Why Not Lower Score:**

None of the reviewers recommends rejection of the paper.

---

### Decision · Program_Chairs · 2024-01-16

Accept (poster)